# IPGO: Indirect Prompt Gradient Optimization for Text-to-Image Model Prompt Finetuning

## Abstract

Text-to-Image (T2I) Diffusion models have become the state-of-the-art for image generation, yet they often fail to align with specific reward criteria such as aesthetics or human preference. We propose **Indirect Prompt Gradient Optimization (IPGO)**, a novel model-independent framework that enhances prompt embeddings by injecting learnable text embeddings as prefix and suffix around the original prompt embeddings. IPGO leverages reduced-rank approximation and rotation, while enforcing range and orthonormality constraints and a conformity penalty to ensure stability and mitigate reward hacking. We evaluate IPGO against six baseline methods under single-prompt optimization with three reward models targeting image-text alignment, image aesthetics, and human preferences across three datasets of varying prompt complexity. The results show that IPGO consistently outperforms all baselines over strong competitors such as DRaFT-1 and TextCraftor. We investigate and mitigate reward hacking, in particular for aesthetics. Ablation studies further highlight the contributions of each IPGO component, while additional experiments demonstrate IPGO's application to various T2I diffusion models.

## 1 Introduction

Text-to-Image (T2I) Diffusion models have emerged as state-of-the-art pipelines for image generation (Liu et al., 2024; Zhang et al., 2023). However, images generated from user prompts often fail to meet specific downstream objectives, such as aesthetic quality or alignment with human preferences (Liu et al., 2024). Further aligning generated images with human evaluations is therefore highly desirable. A number of approaches have been proposed to address this challenge Liu & Chilton (2022); Black et al. (2023); Prabhudesai et al. (2023); Liu et al. (2024); Hao et al. (2024); Li et al. (2024b); Fan et al. (2024); Li et al. (2024b), but they typically rely on data- or computing-intensive training paradigms such as Supervised Fine-Tuning (SFT) and Reinforcement Learning (RL), or require significant modifications to the generative model for each downstream task. As a result, there is continued interest in developing more efficient frameworks for alignment-driven image optimization.

In this paper, we introduce a novel approach, called **I**ndirect **P**rompt **G**radient **O**ptimization (IPGO), which improves image quality by optimizing the prompt itself. Our method is inspired by the linguistic notion of semantic heads or tails – short phrases placed at the beginning or end of a clause to disambiguate, provide context, and add emphasis, intensity, or meaning to it. Analogously, IPGO *injects a few embeddings* at the beginning (prefix) and end (suffix) of the original text prompt embeddings. We employ constrained gradient-based optimization of a rotated reduced-rank approximation to these embeddings to enhance the alignment of the visual representation of the prompt with human judgments through reward guidance. This prefix-suffix tuning strategy offers a modular, model-independent, parameter-efficient approach to prompt optimization, and requires no modifications to the diffusion model or text encoder.

We evaluate IPGO on the Stable Diffusion model (Rombach et al., 2022) using a single L4 GPU with 22.5GB of VRAM. Experiments target three reward models in single-objective optimization settings: (i) image-text alignment (Radford et al., 2021), (ii) image aesthetics (Schuhmann, 2024), and (iii) human preference scores (Wu et al., 2023). The main contributions of this study are:

1. We propose IPGO, a **model-agnostic** gradient-based approach to single-prompt optimization in the text embedding space for reward guidance of T2I diffusion models. This approach op-

timizes **rotated reduced-rank prefix and suffix embeddings** inserted at both the beginning and end of the original prompt embeddings, under range and orthonormality constraints and a conformity penalty.

2. Our extensive experiments on three different datasets and three reward functions, show that for single-prompt optimization, IPGO **consistently outperforms six state-of-the-art methods**, achieving an average **improvement of 1-3% – comparable to gains reported in previous work** (Black et al., 2023; Clark et al., 2023; Hao et al., 2024; Li et al., 2024b). These improvements hold over the strongest benchmarks (TextCraftor and DRaFT-1), while requiring much fewer parameters.

3. Ablation studies highlight the **individual contributions of the constraints** imposed on the optimization, the reduced-rank approximation and rotation, and the length of the prefix and suffix. Furthermore, while we apply IPGO using SDv1.5 for computational efficiency, an additional experiment demonstrates that IPGO is model-independent and can be **applied across a wide range of diffusion models**, including more advanced architectures like SDXL (Podell et al., 2023), SD3 (Esser et al., 2024) and FLUX 1.0-Schnell (BlackForestLabs, 2024).

## 2 RELATED WORK

**Text-to-Image Diffusion Probabilistic Models**   Foundational work in T2I generation using diffusion models includes diffusion-probabilistic models (Sohl-Dickstein et al., 2015), score-based generative models (Song & Ermon, 2019) and the landmark denoising diffusion probabilistic model (DDPM; Ho et al., 2020). Subsequent models, such as GLIDE (Nichol et al., 2021) and Imagen (Saharia et al., 2022) also apply the diffusion process directly in the pixel space. In contrast, methods like Stable Diffusion (Rombach et al., 2022) and DALL-E (Ramesh et al., 2022) apply the diffusion process in a low-dimensional embedding space. Notably, Stable Diffusion has demonstrated superior image quality and efficiency (Zhang et al., 2023), and several extensions to it have been proposed (e.g., Esser et al., 2024; Peebles & Xie, 2023; Podell et al., 2023). FLUX (BlackForestLabs, 2024) is a family of diffusion models that provides high visual quality, fidelity in rendering complex features, and excellent prompt adherence. A key challenge with diffusion models, however, is their potential misalignment with human preferences. A stream of recent work addresses this by controlling models towards preferred properties, either during training or via training-free methods (Liu et al., 2024).

**Training-based alignment**   (Liu et al., 2024) uses supervised fine-tuning (SFT) of the diffusion model combined with reinforcement learning from human feedback (RLHF) to align the model with human preferences, approximated via a reward model. Models in this category, such as ReFL (Xu et al., 2024), DDPO (Black et al., 2023), AlignProp (Prabhudesai et al., 2023), DRaFT (Clark et al., 2023), DPOK (Fan et al., 2024), and DPO-Diffusion (Wang et al., 2024), rely on gradient-based fine-tuning of the diffusion model. Alternatively, models like Diffusion-DPO (Wallace et al., 2024), D3PO (Yang et al., 2024), and SPO (Liang et al., 2024) are directly optimized on preference data. Training-based alignment methods often require considerable computational resources.

**Training-free alignment**   (Liu et al., 2024) aligns diffusion models with human preferences without the need for fine-tuning the diffusion model. The first stream of research uses both manual and systematic approaches to prompt optimization (Oppenlaender, 2023; Wang et al., 2023). Automatic prompt optimization methods, such as Promptist (Hao et al., 2024) and OPT2I (Mañas et al., 2024)), leverage LLMs to refine prompts. The second stream focuses on modifying negative prompts using LLMs (e.g., DPO-Diffusion, Wang et al., 2024) or directly learning negative embeddings (e.g., ReNeg, Li et al., 2024a). The third stream steers the initial noise input of diffusion models towards the reward, such as Best-of-N which generates multiple candidates and selects the one with the highest reward (Nakano et al., 2021), DNO which uses probability-based noise-space regularization (Tang et al., 2024), and ReNO (Eyring et al., 2024) which optimizes the initial noise input for a limited number of steps using a reward model to guide the process. Best-of-N assumes that a high-reward input image exists within a random sample; regularization of the noise distribution is often insufficient to prevent generating of out-of-distribution images (Zhai et al., 2025).

**Alignment through prompt embedding optimization**   includes methods such as PEZ (Wen et al., 2024), which aligns an image with text embeddings of prompts that reflect both the image content

and style. Textual Inversion (Gal et al., 2022) aligns new word tokens with novel objects or styles. TextCraftor (Li et al., 2024b) and TexForce (Chen et al., 2024) align generated images with rewards by fine-tuning the CLIP text encoder within the diffusion pipeline.

Our proposed method **IPGO also optimizes prompt text embeddings**. However, *unlike prompt embedding methods*, such as PEZ (Wen et al., 2024) which requires access to images during prompt discovery, and Textual Inversion (Gal et al., 2022) which requires a few images to learn text embeddings, IPGO operates without accessing such ground-truth images but leverages abstract reward models to guide prompt optimization within the text embedding space. *In contrast to TextCraftor (Li et al., 2024b) and TexForce (Chen et al., 2024)*, which change the text embedding space by fine-tuning the entire text encoder, IPGO explores the embedding space without altering the encoder's parameters, and keeps the original prompt intact, which allows better user control over the prompt's visual representation, and uses much less than one percent of the parameters of these other models. In addition, *counter to adapter-based PEFT approaches such as LoRA (Hu et al., 2021)* which insert trainable parameters in layers of a pre-trained and frozen model, IPGO inserts the trainable prefix and suffix directly into the prompt embeddings, and is thus much more parameter efficient.

In the benchmarking experiments we will demonstrate that IPGO outperforms six benchmark approaches, including PEFT and fully finetuned models, across three datasets and three reward models. *IPGO outperforms its closest competitors TextCraftor and DRaFT-1 in most of the scenarios, but with less than 0.5% of the full parameters, by a significant margin of 1-3%.*

## 3 PRELIMINARIES

**Diffusion Models**  Diffusion models generate images conditioned on a text prompt by sequentially denoising an image from pure Gaussian noise using an error model $\epsilon_\phi$ (Rombach et al., 2022), parameterized by $\phi$. The model $\epsilon_\phi$ predicts the noise in image $x_t$, which is obtained by progressively adding Gaussian noise $\epsilon$ to the original image $x_0$ at each step $t = 0, .., T$ (Ho & Salimans, 2022).

**Reward Models**  Typically, a generated image is evaluated using a pre-trained reward model, $\mathcal{S}$, which assesses by proxy how well the image aligns with human evaluations. For each image $x$ generated by the diffusion model in response to a prompt $p$, the reward model assigns a reward $\mathcal{S}(x, p)$. This reward is then used to guide the diffusion model towards generating images with a higher reward. Widely used reward models are the CLIP loss derived from the multi-modal CLIP model (Radford et al., 2021), the human preference score HPSv2 (Wu et al., 2023), and the LAION aesthetic predictor V2 (Schuhmann, 2024). These models have played a critical role in aligning the outputs of diffusion models with human preferences in research and practice; nonetheless reward hacking (Eyring et al., 2024), where the optimization converges on high-reward images that deviate substantially from the original prompt, can be a significant problem. MIRA (Zhai et al., 2025) uses score-based regularization to prevent the optimized image distribution to deviate significantly from the base image distribution.

## 4 METHODS

Suppose we have a trained reward model $S(x, p)$ on image $x$ and the prompt $p$; a text encoder $\mathcal{T}(\cdot)$ which converts $p$ to its text embeddings $\mathcal{T}(p) \in \mathbb{R}^{d \times K}$, where $d$ is the embedding dimension and $K$ is the length of the tokenized prompt; and a diffusion model characterized by $q_{\text{image}}(x_0 | \mathcal{E}, z_T)$, the probability distribution of the image $x_0$, given prompt-text embeddings $\mathcal{E}$ and a fixed latent state $z_T$ at timestep $T$.

**IPGO**  adds to the original embeddings $\mathcal{T}(p)$ of a text prompt $p$, a prefix $V_{\text{pre}}$ and a suffix $V_{\text{suf}}$, consisting of $N_{\text{pre}}$ and $N_{\text{suf}}$ trainable embeddings, each of dimension $d$, and parameterized by $\Omega_{\text{IPGO}}$. IPGO inserts the prefix at the beginning and the suffix at the end of $\mathcal{T}(p)$, thereby producing an augmented set of text embeddings:

$$\mathcal{E}(V_{\text{pre}}, p, V_{\text{suf}}; \Omega_{\text{IPGO}}) = V_{\text{pre}} \oplus \mathcal{T}(p) \oplus V_{\text{suf}}, \tag{1}$$

where $\mathcal{E}(V_{\text{pre}}, p, V_{\text{suf}}; \Omega_{\text{IPGO}}) \in \mathbb{R}^{d \times (N_{\text{pre}} + K + N_{\text{suf}})}$ and $\oplus$ stands for the concatenation in the second dimension. IPGO optimizes $\Omega_{\text{IPGO}}$ such that the expected rewards of the images sampled from

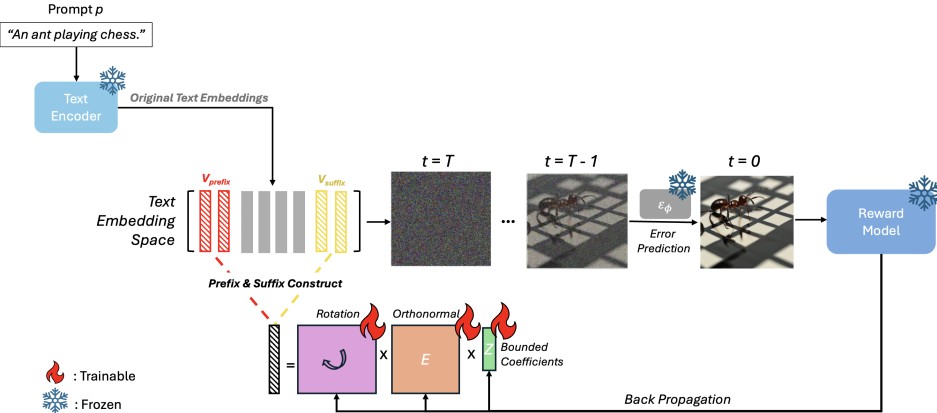

Figure 1: IPGO inserts trainable prefix and suffix embeddings, leveraging reduced-rank approximation and rotation, to the text embeddings of the prompt in the text embedding space, and then sends back reward signals through backpropagation under range and orthonormality constraints, subject to a conformity penalty.

$q_{\text{image}}$ conditioned on a fixed $z_T$ and $\mathcal{E}(V_{\text{pre}}, p, V_{\text{suf}};\ \Omega_{\text{IPGO}})$ are maximized, which is equivalent to minimizing the single-reward loss function:

$$\mathcal{L}(\Omega_{\text{IPGO}}) = -\mathbb{E}_{x_0 \sim q_{\text{image}}(x_0 | \mathcal{E}(V_{\text{pre}}, p, V_{\text{suf}};\ \Omega_{\text{IPGO}}), z_T)} \mathcal{S}(x_0, p). \tag{2}$$

In the following sections, we present the motivation behind our approach and outline the framework. Figure 1 presents a schematic overview of the IPGO methodology.

**Constrained Prefix-Suffix Tuning.** Inspired by Prefix-Tuning (Li & Liang, 2021), IPGO adds extra continuously differentiable embeddings before and after the original text embeddings, as described by equation 1. However, Li & Liang (2021) show that *directly updating prefix embeddings may lead to unstable optimization*. Thus, rather than directly optimizing $V_{\text{pre}}$ and $V_{\text{suf}}$, we reparameterize them as rotated linear combinations of a set of lower-dimensional learnable embeddings. We use a *reduced rank* representation which reduces the number of parameters (Hu et al., 2021), by parameterizing $V_*$ (∗ stands for "pre" or "suf" from here on) as:

$$V_* = \tilde{R}_{2,\theta_2^*} \tilde{R}_{1,\theta_1^*} E_* Z_*, \tag{3}$$

where $E_* \in \mathbb{R}^{d \times m_*}$ is a *trainable set of text embeddings* and $m_*$ is the length of the basis; $Z_* \in \mathbb{R}^{m_* \times N_*}$ are linear coefficients of the basis. The intuition behind equation 3 is that the product $E_* Z_*$ is an $m_*$−dimensional reduced-rank approximation to the original $d$−dimensional embedding, with $m_* \ll d$, using an orthonormal basis $E_*$, with parameters constrained to $Z_* \in [-1, 1]$ (we use $d = 756$ and $m_* = 300$, see Table 8).

**Range constraints.** We impose range constraints on the affine transformation coefficients $Z_* \in [-1, 1]$. These constraints improve training stability and prevent exploding gradients by normalizing the lengths of the prefix and suffix embeddings so that they do not overwhelm the semantics of the original prompt. They thus regularize the optimization process, helping to mitigate reward hacking that occurs because of large coefficient values, by making it harder for the optimization to drift towards high-reward out-of-distribution regions that are not well aligned with the prompt (Shihab et al., 2025).

**Orthonormality constraint.** We impose an orthonormality constraint on the text embedding basis $E_*$, i.e., $E_* E_*^T = I_{m_*}$, to ensure each dimension of the basis represents a semantic concept that is disentangled from others (Jiang et al., 2023), thus preventing semantic leakage across dimensions (Sharifdeen et al., 2025). Orthonormality standardizes the subspace basis and ensures maximal subspace size.

**Rotation.** We apply pairwise rotations to the $d$ dimensions of $E_* Z_*$ via orthogonal rotation matrices $R_{1,\theta_1}$ and $R_{2,\theta_2}$. The basis $E_*$ helps explore a subspace of the text embedding space that is rotated to fit with the reward guidance. The embeddings are rotated pairwise in two-dimensional subspaces which preserves the embedding vector angles and norms, thus preserving the relative importance of the embedded semantic concepts, while aligning the prefix and suffix embeddings with the reward (Hamilton et al., 2016). We use the fact that a full orthogonal rotation matrix can be factored into a product of independent pairwise rotations (Givens, 1958; Liang et al., 2025). Exploring the orthogonal subspace along with rotations is more efficient than exploring the original embeddings (see Appendix C); exploring pairwise rotations is more efficient than exploring the full $d$-dimensional rotation. The pairwise rotations not only introduce non-linearity, but they accelerate the search process via their gradient directions (see Appendix C).

We apply two rotation matrices $\tilde{R}_{1,\theta_1^*}$ and $\tilde{R}_{2,\theta_2^*}$. The rotation matrices are composed of the $2 \times 2$ elementary matrix controlled by angle $\theta \in (-\frac{\pi}{2}, \frac{\pi}{2}]$:

$$R_{e,\theta} = \begin{bmatrix} \cos\theta & -\sin\theta \\ \sin\theta & \cos\theta \end{bmatrix}. \tag{4}$$

Given $\theta_1^*$ and $\theta_2^*$, we define $\tilde{R}_{1,\theta_1^*} \in \mathbb{R}^{d \times d}$ and $\tilde{R}_{2,\theta_2^*} \in \mathbb{R}^{d \times d}$ by:

$$\tilde{R}_{1,\theta_1^*} = I_{d/2} \otimes R_{e,\theta_1^*}, \quad \tilde{R}_{2,\theta_2^*} = \begin{bmatrix} R_{e,\theta_2^*,(2)} & & \\ & I_{d/2-1} \otimes R_{e,\theta_2^*} & \\ & & R_{e,\theta_2^*,(1)} \end{bmatrix}, \tag{5}$$

where $\otimes$ is the tensor product, $I_a$ is the identity matrix of size $a$, $R_{e,\theta_2^*,(i)}$ is the $i^{th}$ row of the elementary rotation matrix $R_{e,\theta_2^*}$. To interpret, $\tilde{R}_{1,\theta_1^*}$ rotates pairs $(2j-1, 2j)$ and $\tilde{R}_{2,\theta_2^*}$ rotates pairs $(2j, 2j+1)$ of the coordinates of the embedding $E_* Z_*$, where $j = 1, \ldots, d/2$ and the $(d+1)^{th}$ coordinate is the $1^{st}$ coordinate.

**Conformity penalty.** We add a conformity penalty to ensure that the mean of the IPGO embeddings (equation 1) are about the same as the mean of the base prompt: Mean $(\mathcal{T}(p))$ = Mean $(V_{\mathrm{pre}} \oplus \mathcal{T}(p) \oplus V_{\mathrm{suf}})$, promoting coherence of the generated prefix and suffix embeddings with the base prompt. This penalty term is motivated by the observation that the prompt semantics is well reflected in the mean of the original text embeddings (see Appendix D). Note that this penalty is the same as a KL-divergence regularization of the embeddings under the assumption of Gaussians with identity covariance matrices: $D_{KL} = \frac{1}{2} \left(\mathrm{Mean}\left(\mathcal{T}(p)\right) - \mathrm{Mean}\left(V_{\mathrm{pre}} \oplus \mathcal{T}(p) \oplus V_{\mathrm{suf}}\right)\right)^2$. We use a default penalty weight of $\gamma = 0.001$. This penalty thus acts as a regularizer to promote coherence of the generated prefix and suffix embeddings with that of the base prompt, thus avoiding over-optimization of the pre- and suffix embeddings, which depending on the value of $\gamma$, helps to mitigate reward hacking (Khalaf et al., 2025).

## 5 EXPERIMENTS AND RESULTS

We conduct a set of experiments to evaluate the performance of IPGO across six benchmark models on three datasets. In Section 5.1, we describe the experiment settings, while Section 5.2 introduces the benchmarks, and Section 5.3 presents the results.

### 5.1 EXPERIMENT SETTINGS

**Datasets.** Three datasets are considered: the COCO image captions (Lin et al., 2014), DiffusionDB (Wang et al., 2022), and Pick-a-Pic (Kirstain et al., 2023). These datasets represent a wide range of prompts and images with varying levels of complexity. To assess the performance of IPGO across different categories of image captions, we conduct separate evaluations for COCO images in the following five categories: Persons, Rooms, Vehicles, Natural Scenes, and Buildings. For each category, we randomly select 60 captions. Additionally, we randomly select 300 prompts from both the DiffusionDB and Pick-a-Pic datasets, resulting in a total of 900 prompts for evaluation. The number of prompts in our experiments substantially exceeds those used in recent experiments, such as (Black et al., 2023) and (Wang et al., 2024), both of which relied on approximately 600 prompts.

**Training.**    All experiments with IPGO, which has a total of 0.47M parameters, are conducted on a single NVIDIA L4 GPU with 22.5GB of memory. IPGO takes at most 12GB of memory for all tasks. The backbone diffusion model used is Stable-Diffusion (SD)v-1.5, chosen for its balance between generation quality and computational efficiency (Li et al., 2024b; Podell et al., 2023). Note that IPGO is model-agnostic and directly applicable to other diffusion models as well (as shown in Section 6.3). For comparability, all models and experiments are run in identical computational environments. In contrast, the benchmark TextCraftor (introduced below) requires a single A100 GPU.

**Reward Models.**    To ensure the flexibility and broad applicability of IPGO, we consider publicly available reward models. Specifically, we use the CLIP loss from the multimodal CLIP model (Radford et al., 2021), the human preference score v2 (Wu et al., 2023), and the LAION aesthetic predictor v2 (Schuhmann, 2024). These widely used reward models capture a broad spectrum of criteria, effectively representing the diverse rewards relevant to text-image alignment tasks. In addition, to cross-validate the reward scores of the optimized images, we use VQAScore (Lin et al., 2024), which measures prompt-image alignment based on a visual question answering(VQA) model.

## 5.2    Benchmarks

We evaluate IPGO using the following six benchmarks, which represent the current state of the art (SOTA) in the categories discussed in Section 2. The first baseline is **Stable diffusion with a raw prompt** (Rombach et al., 2022), against which we expect IPGO to enhance performance across all datasets and reward models. The second baseline is **TextCraftor** (Li et al., 2024b), using a fine-tunable text encoder with 123M parameters, representing the current SOTA among text-embedding-based methods. We also use two training-based methods: **DRaFT** (Clark et al., 2023) and **DDPO** (Black et al., 2023). For DRaFT, we select the DRaFT-1 variant with LoRA of rank 3 as the baseline (#parameters: 0.60M), due to its low computational cost and competitive performance (Clark et al., 2023). For DDPO (#parameters: 0.79M), we apply the default LoRA configuration. Although these models are training-based and offer inference after a one-time training cost, we believe that a comparison with them on focused single-prompt training where no generalization is required, is of interest. Moreover, we include two training-free methods: **DPO-Diff** (Wang et al., 2024), and **Promptist** (Hao et al., 2024), which are single-prompt methods and therefore the key competitors to IPGO. Promptist can be used for multi-objective optimization, but is applied to single-objective optimization here. Detailed qualitative comparisons between several benchmarks and IPGO can be found in Appendix E. The hyperparameter settings are provided in Table 8 in Appendix F.

## 5.3    Single-Prompt Image Optimization

We train all methods listed in Table 7, using a single prompt at a time. Single image optimization during inference is more flexible and facilitates generalization to unseen prompts (Eyring et al., 2024). For all six benchmarks we use default configurations for learning and sampling. The best loss value achieved during training is used to represent the final performance of each method. In addition to comparing the absolute loss, we compute the *percentage improvements* IPGO gains (in parentheses) over the best baseline. We also report the $t-$ *statistics and* $p-$ *values* of the overall average improvements of IPGO over the best baseline on all three rewards.

**Alignment.**    Table 1 presents the results of IPGO and benchmark methods for CLIP semantic alignment across three datasets. With the CLIP reward, IPGO outperforms all six benchmarks in all scenarios except for COCO-Buildings. Note strong alignment for COCO-Person prompts in particular, and for the more complex prompts from DiffusionDB. IPGO achieves the highest average alignment scores across all prompts, surpassing the top benchmark DRaFT-1 by 1.8% ($t$-value= 3.9, $p = 4\mathrm{e}{-}05$). It improves alignment by 17.2% over the original SDv1.5 diffusion model ($t$-value = 23.8, $p < 1\mathrm{e}{-}10$).

**Preferences.**    Table 2 presents the results for HPSv2 human preference scores. Again, IPGO outperforms all benchmarks on all datasets; its highest preference scores are achieved for COCO-Person and COCO-Vehicle prompts. IPGO's average reward score across all datasets is the highest, achieving an average improvement of 1.0% ($t$-value= 1.7, $p = 0.046$) over the strongest benchmark TextCraftor, where the smaller effect size and statistical significance is caused by heterogeneity in

| Dataset | IPGO (↑) | SD v1.5 | TextCraftor | DRaFT-1 | DDPO | DPO-Diff | Promptist |
|---|---|---|---|---|---|---|---|
| COCO | | | | | | | |
|   Person | **0.3160** (2.4) | 0.2637 | 0.3085 | 0.3067 | 0.2865 | 0.2911 | 0.2598 |
|   Room | **0.2883** (3.5) | 0.2482 | 0.2786 | 0.2782 | 0.2648 | 0.2755 | 0.2398 |
|   Vehicle | **0.2986** (1.8) | 0.2514 | 0.2928 | 0.2934 | 0.2755 | 0.2881 | 0.2474 |
|   Natural Scenes | **0.2922** (1.6) | 0.2539 | 0.2876 | 0.2802 | 0.2614 | 0.2815 | 0.2307 |
|   Buildings | 0.2846 (-1.8) | 0.2439 | 0.2859 | **0.2898** | 0.2718 | 0.2794 | 0.2377 |
| DiffusionDB | **0.3247** (2.5) | 0.2759 | 0.3146 | 0.3167 | 0.3024 | 0.2929 | 0.2753 |
| Pick-a-Pic | **0.3125** (0.2) | 0.2681 | 0.3077 | 0.3103 | 0.2946 | 0.2980 | 0.2612 |
| Avg. Reward | **0.3110** (1.8) | 0.2654 | 0.3041 | 0.3056 | 0.2897 | 0.2913 | 0.2599 |

Table 1: Comparison of IPGO's **CLIP alignment scores** with six benchmarks across 900 prompts from three datasets. Bold/underline denote highest/second-highest scores. In parentheses are percentage improvements over the second-best performing model, DRaFT-1.

HPSv2 scores across images, and an improvement of 6.0% ($t$-value=17.4, $p < 1e-10$) over the original SDv1.5 model.

| Dataset | IPGO (↑) | SD v1.5 | TextCraftor | DRaFT-1 | DDPO | DPO-Diff | Promptist |
|---|---|---|---|---|---|---|---|
| COCO | | | | | | | |
|   Person | **0.2950** (1.4) | 0.2796 | 0.2905 | 0.2786 | 0.2819 | 0.2481 | 0.2680 |
|   Room | **0.2817** (0.4) | 0.2673 | 0.2806 | 0.2646 | 0.2711 | 0.2430 | 0.2596 |
|   Vehicle | **0.2917** (0.4) | 0.2761 | 0.2905 | 0.2755 | 0.2814 | 0.2491 | 0.2679 |
|   Natural Scenes | **0.2866** (0.6) | 0.2721 | 0.2848 | 0.2667 | 0.2741 | 0.2487 | 0.2600 |
|   Buildings | 0.2867 (-0.5) | 0.2719 | **0.2882** | 0.2723 | 0.2782 | 0.2580 | 0.2634 |
| DiffusionDB | **0.2729** (0.4) | 0.2594 | 0.2719 | 0.2602 | 0.2634 | 0.2381 | 0.2585 |
| Pick-a-Pic | **0.2753** (0.6) | 0.2621 | 0.2741 | 0.2647 | 0.2672 | 0.2509 | 0.2591 |
| Avg. Reward | **0.2788** (0.5) | 0.2650 | 0.2776 | 0.2655 | 0.2693 | 0.2461 | 0.2605 |

Table 2: Comparison of IPGO's **HPSv2 human preference scores** with six benchmarks across 900 prompts from three datasets. Bold/underline denote highest/second-highest scores. In parentheses percentage improvements over the second-best performing model, TextCraftor.

**Aesthetics.** Table 3 presents the results for LAION aesthetics scores. IPGO outperforms all benchmarks across every dataset. Aesthetics scores are particularly high for the COCO-Person, Pick-a-Pick and DiffusionDB prompts. IPGO's average reward score is the highest across all datasets, with an improvement of 3.2% ($t$-value= 6.5, $p = 5e-10$) over the best benchmark, TextCraftor, and an improvement of 16.5% ($t$-value= 34.0, $p < 1e-10$) over the original SDv1.5 model.

| Dataset | IPGO (↑) | SD v1.5 | TextCraftor | DRaFT | DDPO | DPO-Diff | Promptist |
|---|---|---|---|---|---|---|---|
| COCO | | | | | | | |
|   Person | **6.2174** (4.7) | 5.2447 | 5.8365 | 5.7761 | 5.5777 | 4.2865 | 5.9401 |
|   Room | **5.7549** (2.8) | 5.0931 | 5.5994 | 5.4426 | 5.3700 | 4.1589 | 5.5993 |
|   Vehicle | **5.8567** (3.3) | 4.9608 | 5.6699 | 5.5063 | 5.4219 | 4.0197 | 5.5643 |
|   Natural Scenes | **5.9301** (3.3) | 5.0558 | 5.7436 | 5.6156 | 5.5099 | 4.2952 | 5.6483 |
|   Buildings | **5.7987** (1.9) | 5.0326 | 5.6909 | 5.4294 | 5.3484 | 4.2777 | 5.6431 |
| DiffusionDB | **6.3469** (0.2) | 5.5012 | 6.3318 | 6.1100 | 5.9644 | 4.4350 | 5.6291 |
| Pick-a-Pic | **6.2684** (4.5) | 5.3289 | 5.9978 | 5.9048 | 5.7547 | 4.3565 | 5.6954 |
| Avg. Reward | **6.1735** (2.7) | 5.3025 | 6.0117 | 5.8563 | 5.72156 | 4.3330 | 5.6678 |

Table 3: Comparison of IPGO's **LAION-Aesthetics scores** with six benchmarks across 900 prompts from three datasets. Bold/underline denote highest/second-highest scores. In parentheses percentage improvements over the second-best performing model, TextCraftor.

**Reward hacking.** To investigate if IPGO is subject to reward hacking, we cross-validated the results with the VQAScore for prompt-image alignment (Lin et al., 2024). Reward hacking is evidenced by a significant drop in the VQAScore prompt-alignment of the optimized images relative to the original images produced with SDv1.5 from the base prompt. Table 4 presents these scores for the original images, and for the images guided towards CLIP, human preference, and aesthetics

rewards, respectively. Images steered towards higher CLIP and human preference rewards have higher VQAScores than the original images, and thus show no evidence of reward hacking. However, for the LAION aesthetics scores, the VQAScores are substantially lower than those for SDv1.5, which shows that IPGO over-optimizes the aesthetics reward at the expense of the alignment of the image with the base prompt. It is widely known that LAION-Aesthetics model is prone to reward hacking because it lacks internal semantics supervision (Clark et al., 2023; Li et al., 2024b; Zhai et al., 2025). Visual inspection of the optimized images confirms that reward hacking in some cases produces out-of-distribution images; Appendix I shows two examples. Appendix J illustrates that the problem of reward hacking can be exacerbated by over-parametrization, that is very long prefix and suffix relative to the base prompt. We propose two methods to mitigate reward hacking.

①As an **in-training** method for mitigating hacking of the LAION aesthetics reward, we increase the conformity penalty weight to $\gamma = 70$. Such large penalty is chosen to (1) match the scale of aesthetics scores and (2) overcome the LAION-Aesthetics model's intrinsic absence of semantics supervision. With the large penalty, the average aesthetics scores for COCO, DiffusionDB, and PickaPic datasets improve over SDv1.5 from 5.03, 5.50, and 5.32, to 5.30, 5.70, and 5.60, respectively. The average VQAScores with larger penalty are similar to those of SDv1.5 and much better than those of IPGO without the default penalty, which reveals that this strategy helps mitigate reward hacking. Appendix I illustrates the effects of increasing the penalty for two images.

②As a **post-training method** to mitigate hacking of the aesthetics reward, we blend IPGO's prefix/suffix embeddings for CLIP and aesthetics scores post-training (based on a grid search we use a 0.2/0.8 ratio [1]) and apply Best-of-5 with respect to the CLIP score. Such blending can provide some semantics while generating high aesthetic images. Table 4 shows that this strategy leads to much better VQAScore alignment than the SDv1.5 model and thus mitigates reward hacking: for example, with this post-processing the average aesthetics score on the COCO prompts is 5.43, much higher than 5.08 using the base prompt and SDv1.5, while the VQAScore increases from 0.741 to 0.786.

| Dataset | SD v1.5 | IPGO-CLIP | IPGO-HPSv2 | IPGO-LAION | IPGO-Penalty | IPGO-Blend |
|---|---|---|---|---|---|---|
| COCO | 0.741 | 0.790 | 0.773 | 0.575 | 0.727 | 0.786 |
| DiffusionDB | 0.716 | 0.761 | 0.722 | 0.615 | 0.710 | 0.745 |
| Pick-a-Pic | 0.650 | 0.714 | 0.689 | 0.594 | 0.654 | 0.702 |

Table 4: Comparison of **VQAScores** for images from three datasets optimized for respectively CLIP reward, HPSv2 human preference reward, and LAION aesthetics reward, with original images generated with Stable Diffusion SD v1.5. IPGO-blend blends IPGO's prefix/suffix embeddings for CLIP and aesthetics scores (with 0.2/0.8 weights) post-training to mitigate reward hacking. IPGO-Penalty increases the conformity penalty in training to mitigate reward hacking.

**Qualitative interpretation.** Figure 2 qualitatively compares a non-cherry-picked sample of images generated with IPGO using the HPSv2 reward to those generated with the raw prompt and with TextCraftor and DRaFT-1, the best performing benchmarks (note that the computational environment affects the quality of each image in Figure 2 equally). Unlike DRaFT-1 and TextCraftor, which often drastically alter the image layout from the one produced by the base prompt, IPGO tends to modify or add details, while preserving the layout produced with the base prompt, thus providing enhanced control over image generation. Additional examples can be found in Appendix K.

**Summary.** Across all 126 (6 baselines × 7 scenarios × 3 rewards) comparisons, *IPGO yields the best performance in all but 2 of the cases*, yielding a *significant 6-17% improvement over the raw prompt*, and a *significant 1-3% improvement* over the best-performing benchmarks, across three rewards, which is similar in magnitude to improvements reported for prior models (e.g., Black et al., 2023; Clark et al., 2023; Hao et al., 2024; Li et al., 2024b), while retaining the global image layout obtained from the base prompt. This holds in particular for prompt alignment and human preference rewards, but for the aesthetic rewards cross-validation with the VQAScore reveals evidence of reward hacking. Average clock-times for training are between 3-5 minutes per image, on a single NVIDIA L4 GPU. *Viable strategies to mitigate reward hacking are increasing the weight on the conformity penalty, or blending the trained prefix/suffix embeddings obtained for CLIP and aesthetics rewards.*

---

[1]We performed a grid search on ratios (0.1,0.9), (0.2,0.8) and (0.3,0.7).

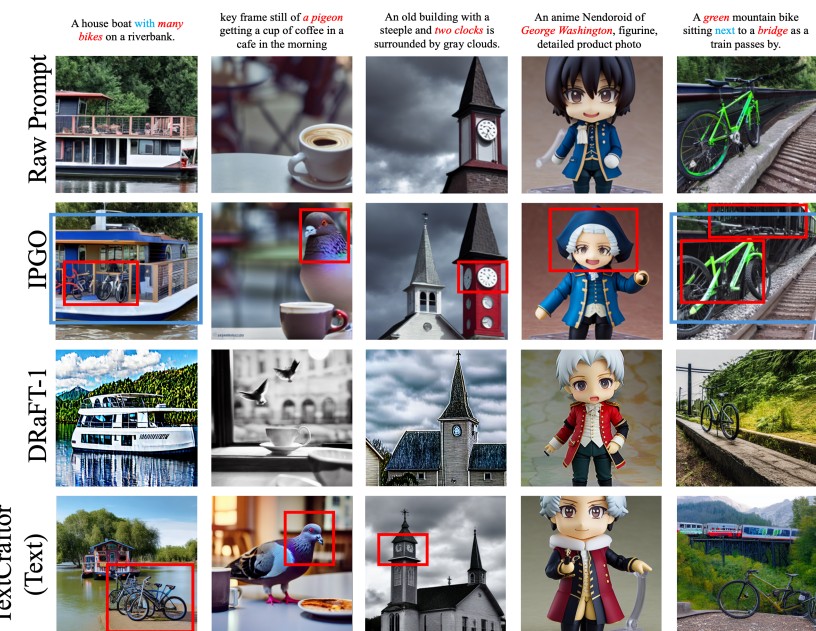

Figure 2: Example images generated with Stable Diffusion v1.5 using the raw prompt (row 1), IPGO (row 2), DRaFT-1 (row 3) and TextCraftor (row 4), towards the HPSv2 reward. Unlike DRaFT-1 and TextCraftor, IPGO tends to modify or add details providing better prompt alignment with a disentangled HPSv2 reward, while preserving the layout produced with the original prompt.

# 6 FURTHER EXPERIMENTS

## 6.1 ABLATION STUDIES

We provide in-depth ablation studies on the components of our IPGO framework, including the constraints, penalty, rotation, and the lengths of the prefix and suffix. All experiments are conducted on the full COCO dataset using 300 prompts across three rewards. The Stable Diffusion pipeline SDv1.5 is configured with 30 inference steps, and all models are trained for up to 30 epochs. More ablation studies are covered in Appendix G.

**Effects of the Constraints, Rotation, and Penalty.** We evaluate the effects of the Range ($R$) and Orthonormality ($O$) constraints, the Conformity ($C$) penalty (with default weight $\gamma = 0.001$), as well as the rotation component, on the performance of IPGO. Specifically, we compare six scenarios, the full IPGO model that incorporates all constraints, three versions of IPGO where each of the $O$, $C$, or $R$ constraints is omitted, IPGO without the rotation component, and finally, the IPGO without any constraints, the reduced-rank design, or rotations (thus directly fine-tuning the prefix and suffix text embeddings). Table 5 presents the results.

| Scenarios | Aesthetics | Alignment | Human Preference |
|---|---|---|---|
| Full IPGO | **6.0626** | 0.2771 | **0.2766** |
| w/o Orthonormality constraint | 5.9892 | 0.2709 | 0.2733 |
| w/o Conformity penalty | 5.9422 | 0.2767 | 0.2763 |
| w/o Range constraint | 5.9151 | 0.2770 | 0.2703 |
| w/o Rotation | 5.9247 | **0.2779** | 0.2762 |
| Naïve learning w/o constraints, penalty & rotation | 5.1462 | 0.2442 | 0.2630 |

Table 5: Ablation experiments to test the effects of penalty, constraints ($O$, $C$, $R$) and rotation, relative to the Full IPGO model, and a naïve full rank model.

First, the *Full IPGO consistently performs best when all of the components are included*. Each reward benefits most from a specific optimization constraint or penalty. For aesthetics and human preference rewards, the range ($R$) constraint yields the most performance gains. However, for the CLIP alignment

score, orthonormality (*O*) is the most important. Therefore, although each component's contribution towards the final solutions depends on the reward model, *IPGO with all three components combined adapts to different reward tasks to each deliver consistent image alignment*. Second, the *effects of the rotation parametrization also depend on the reward model*: it helps aesthetics scores to improve the most (2.3%), then the human preference scores (0.1%), but it not necessarily improves alignment scores. The conformity penalty already induces alignment, which may make the rotation redundant for the aligning the semantic meaning in the prefix and suffix embeddings with the base prompt. The benefits of rotation are further explained in Appendix C. Third, *the Full IPGO outperforms the naive IPGO without any constraints, penalty, and parameterization by a large margin* (17.8%, 13.5%, and 5.2% on aesthetics, alignment and human preference scores), showing the effectiveness of IPGO's optimization over naïve, unconstrained text embedding learning.

## 6.2 TRANSFER ACROSS INITIAL NOISE DISTRIBUTIONS

A powerful method for reward guidance of T2I image generation is to is to steer the initial noise input of diffusion models towards the reward, through methods such as Best-of-N (Nakano et al., 2021). In the previous experiments we fixed the seed and used a single noise-input distribution to enable a fair comparison across methods. We conduct an experiment to test whether IPGO's prefix/suffix embeddings are transferable to different random seeds, and compare that with Best-of-N, with $N = 30$ for all COCO prompts. We obtain a CLIP score of 0.295 for Best-of-N, versus 0.301 for IPGO with embeddings transferred across initial noise distributions. For human preference, Best-of-N and IPGO with prefix/suffix embeddings transferred across noise-input distributions yield the same score, 0.289. Our initial conclusion from this experiment is that, with embedding transfer across initial noise, IPGO performs at least as good as Best-of-N, and that the trained prefix/suffix embeddings transfer well across initial noise distributions.

## 6.3 ADAPTIVITY OF IPGO TO OTHER DIFFUSION MODELS

We next illustrate that IPGO can be used with different diffusion models. In the experiments reported heretofore we choose to implement IPGO with SD-v1.5, and here we illustrate IPGO, along with TextCraftor as a benchmark, for two newer versions of Stable Diffusion, SDXL Podell et al. (2023) and SD3 Esser et al. (2024), and one much larger and more recent T2I model FLUX 1.0-Schnell BlackForestLabs (2024), for HPSv2 human preference rewards on 100 randomly selected prompts from the DiffusionDB data. Images are sampled with 30 steps of inference; optimizations take 30 epochs. For IPGO, we insert prefix and suffix in the CLIP text encoder for SDXL and SD3, and in the T5v1.1-XXL text encoder Raffel et al. (2020) for FLUX 1.0-Schnell. For Textcraftor, we only fine-tune the CLIP text encoder due to computation limits.

Table 6 shows that *IPGO improves human preference scores over the original prompt for the SD3 (3.2%), SDXL (4.9%) and FLUX 1.0-Schnell (1.8%) diffusion models as well*. Its performance is as good as or better than TextCraftor for these diffusion models. The results demonstrate that IPGO is model independent and can be used with various different T2I diffusion models. In addition, IPGO can optimize very long prompts (0ver 150 words) as well, as shown in Appendix H.

| Diffusion model | SDXL | SD3 | FLUX 1.0-Schnell |
|---|---|---|---|
| Original | 0.2523 | 0.2625 | 0.2609 |
| TextCraftor | 0.2626 | 0.2686 | **0.2657** |
| IPGO | **0.2646** | **0.2710** | 0.2657 |

Table 6: HPSv2 reward for IPGO and TextCraftor on different large Diffusion Models for 100 randomly selected prompts from the DiffusionDB dataset.

## 7 CONCLUSION

IPGO is a model-agnostic, gradient-based prompt-level optimization framework for alignment of generated images with prompt semantics, human preferences, and aesthetics. Extensive experiments over six benchmarks across three tasks, three datasets, and various diffusion model backbones demonstrate IPGO's performance gains on single-prompt optimizations. We leave generalization to different alignment tasks, more reward hacking preventions, batch training, multi-criterion optimization, and interpreting the optimized prefix and suffix embeddings, as topics for future research.

ETHICS STATEMENT

This work adheres to the ICLR Code of Ethics. All datasets used are publicly available and have been cited appropriately. No personally identifiable or sensitive information is included. We have taken care to ensure that our methods and results do not introduce or propagate harmful biases beyond those already present in standard benchmark datasets. The potential societal impacts of this research, both positive and negative, are discussed in Appendix A of the paper.

REPRODUCIBILITY STATEMENT

We have taken several measures to ensure the reproducibility of our results. A complete description of our model architecture and training procedure is provided in Section 4 of the main paper, with further implementation details in Appendix F. The datasets used in our experiments are described in Section 5. An anonymous Git repository containing the source code and scripts will be made available during the discussion phase to facilitate reproducibility.

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

# Appendices

## A    SOCIETAL IMPACT

This paper presents work that contributes to the field of Text-to-Image generation models and their applications. In the Machine Learning community, the new method introduced by this paper can broaden the current horizon on fine-tuning diffusion models. In practice, our method can be applied to image related tasks such as automatic real-time image editing.

## B    LLM USAGE

Large Language Models (LLMs) were used in limited capacities to support this work. Specifically, they were only employed for grammar correction and language polishing during manuscript preparation. Additionally, LLMs were used to assist in additional ablation studies in Appendix G by generating diverse image prompts. No parts of the scientific analysis, experimental design, or core findings relied on LLM outputs.

## C    OPTIMIZATION OF ROTATION PARAMETERIZATIONS

We show the intuition how the rotations help accelerate the optimization process. First, we consider the optimization in the two-dimensional space. Suppose we have a minimization problem

$$\operatorname{argmin}_x f(x), \quad x \in \mathbb{R}^2. \tag{6}$$

We assume this problem only has one global minimum $x^*$, which therefore lies in the subspace spanned by itself. Now we parameterize $x = R_{e,\theta} y$, with $y \in \mathbb{R}^2$ and $R_{e,\theta}$ the elementary rotation matrix in equation 4. We update parameters step by step. We initialize $x_0$ by $\theta_0 = 0$ and $y_0 = 0$ at the origin. A gradient update step moves $x_0$ along the gradient of $x_0$, with a suitable step size, to $x_1$, with $\theta_0$ unchanged. Assume we are currently at $x_t = R_{e,\theta_t} y_t$. We update $\theta_t$ by solving $\theta_{t+1}$ from:

$$\nabla_x f(x_t)^T \frac{dR}{d\theta}\bigg|_{\theta_{t+1}} y_t = 0. \tag{7}$$

Note $\frac{dR}{d\theta}\big|_{\theta_{t+1}} = R_{e,\frac{\pi}{2}} R_{e,\theta_{t+1}}$, the elementary rotation matrix with angle $\theta_{t+1}$ rotated 90 degrees counterclockwise. Therefore, graphically, the optimal $\theta_{t+1}$ is the one that rotates $y_t$, with the origin as the rotation axis, to a point such that the vector pointing to it is parallel to the gradient at that point. In other words, this is the point where the circle with radius $\|x_t\|$ is tangent to the contour of $f$. Then for any suitable step sizes along the corrected gradient towards the new point $x_{t+1}$, the total path length between the origin $x_0$ and the optimal point $x_*$ is equal to the distance $\|x_*\|_2$, which is the shortest path between the initial point and the optimal point, and therefore **optimal** among all possible paths between the initial point to the optimum point. Figure 3 visualizes the argument. The left panel shows an optimization path with rotation, which makes the total path length be exactly equal to the shortest path (the purple line) since the updated points are selected at the tangent point between the circles (red and dashed) and the contour. However, there is no guarantee that a regular gradient descent takes that shortest path, as illustrated on the right panel.

*In a high-dimensional space*, since IPGO rotates each neighboring pair of coordinates, the high-dimensional rotation can be disassembled into separate 2-dimensional rotations. Therefore, our argument above in the 2D space can be extended to the high-dimensional space. In other words, the high-dimensional rotation should guide a relatively shorter optimization path towards the optimal solution. Nonetheless, one difference should be noted. In our IPGO algorithm, because the rotation angles to be optimized are shared across all neighboring pairs of coordinates, overall the rotation component of the IPGO will select the angle that on average benefits the optimization path the most. The optimization along the average path could lead to less efficient optimization updates in some of the 2D subspaces, which may therefore require more optimization updating steps for convergence. This effect can be observed in our ablation studies on rotation for CLIP alignment, shown in Table 5.

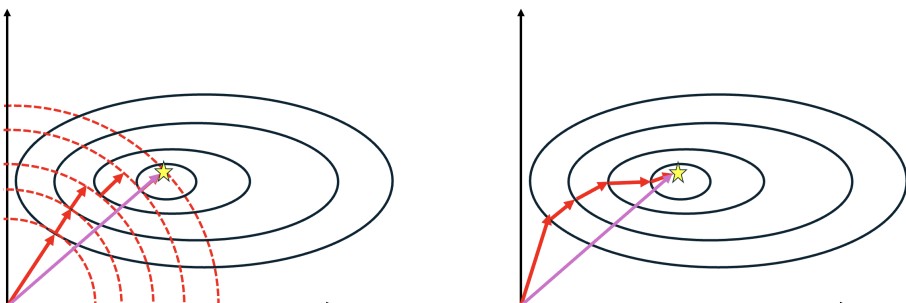

Figure 3: The figure compares the optimizations with rotation parametrization (left) and without (right). The yellow star represents the optimum point. The red dashed lines on the left are the circles with radius of the length of the current $x_t$. The purple line is the shortest distance between the initial point, the origin, and the optimum point. With rotation parametrization, the updates must be along the shortest total path.

## D    CONFORMITY PENALTY FOR PROMPT SEMANTICS PRESERVATION

To investigate if the mean of the text embeddings of the prompts represent the semantics well, we plot the t-SNE-2D-projected vectors of the mean text embeddings for the COCO prompts in Figure 4. The COCO prompts fall in five different categories: 1.*Person*, 2. *Room*, 3. *Vehicle*, 4. *Natural Scenes* and 5. *Buildings*. We observe that even in the 2D projection, almost every category forms its own semantic group. This observation motivates us to impose a penalty on the mean text embedding to prevent it from deviating too far from that or the base prompt and maintain the prompt semantics during the optimization.

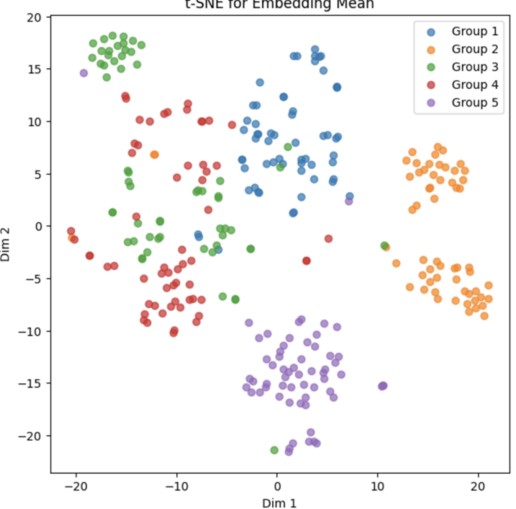

Figure 4: t-SNE Projections of the mean text embeddings. The mean text embeddings reflect most of the image semantics in the 2D projection, as the five different semantic groups in the COCO prompts are mostly well-separated.

# E    COMPARISONS BETWEEN IPGO AND BASELINES

Table 7 qualitatively compares our IPGO algorithm with the benchmarking algorithms outlined in Section 5.2. As can be seen from the table, IPGO is the only method that leverages reward gradient computation, supports prompt modification, and has a relatively low number of parameters.

| Algorithms | Reward Gradient | Prompt Modification | Dimensionality of Parameter Space |
|---|---|---|---|
| Promptist | ✗ | ✓ | High (SFT required) |
| DPO-Diff | ✓ | ✓ | High (External LLM required) |
| TextCraftor | ✓ | ✓ | High (Text Encoder Fine-Tuning) |
| DRaFT | ✓ | ✗ | Low |
| DDPO | ✗ | ✗ | High (Many samples required) |
| IPGO (ours) | ✓ | ✓ | Low |

Table 7: A qualitative comparison between IPGO and all baselines, focusing on three key aspects: the ability to compute reward gradients, support for prompt modification, and the dimensionality of the parameter space.

# F    IMPLEMENTATION DETAILS

This section provides details on IPGO's implementation and training.

**Image Generation.**    A DDIM Scheduler with a guidance weight of 7.5 is employed and the generated images have a resolution of $512 \times 512$ pixels. During optimization, IPGO truncates the backpropagation at the 2nd-to-last sampling step. we set the number of inference step for image generation as 50. We found similar performances with other sampling strategies, such as PNDM and LMSD.

**Optimization**    We train IPGO using Adam (Kingma, 2014) optimizer without a weight decay. We start with a learning rate of $1e-3$ and reduce it by a factor of $0.9$ every 10 epochs, continuing this schedule for a total of 50 epochs. We truncate gradients at the 2nd-to-last step with checkpointing. We apply gradient clipping across all our experiments, selecting a gradient clipping norm of $c = 1.0$.

**Hyperparameters.**    We set the hyperparameters of IPGO, DDPO, and DRaFT-1 to ensure that the total number of trainable parameters is comparable. IPGO includes the hyperparameters: $m_{\mathrm{pre}}(m_{\mathrm{suf}})$, the number of the base text embeddings for prefix (suffix); and $N_{\mathrm{pre}}(N_{\mathrm{suf}})$, the length of the prefix (suffix). DRaFT-1 uses the LoRA parameters in the UNet. For DDPO and TextCraftor we use the default configurations. Detailed hyperparameter settings for IPGO, TextCraftor, DRaFT-1 and DDPO are provided in Table 8.

| Methods | Hyperparameter | Value |
|---|---|---|
| IPGO | $m_{\mathrm{pre}}, m_{\mathrm{suf}}$ | 300 |
|  | $N_{\mathrm{pre}}, N_{\mathrm{suf}}$ | 10 |
| Total #parameters |  | 0.47M |
| TextCraftor | Default Configuration |  |
| Total #parameters |  | 123M |
| DRaFT-1 | LoRA rank | 3 |
| Total #parameters |  | 0.60M |
| DDPO | Default DDPO Trainer Configuration |  |
| Total #parameters |  | 0.79M |

Table 8: Hyperparameter settings for IPGO, TextCraftor, DRaFT-1 and DDPO

**IPGO's Constraints.** IPGO has two constraints: Range and Orthonormality, and a Conformity penalty. We enforce the orthonormality constraint with `orthogonal()` module in Pytorch. For the Range constraint, we clamp the parameters to satisfy the constraint after each update. Finally, we add a conformity penalty to the objective, the negative image reward. Define the conformity penalty by

$$P_{\text{conf}} = \|mean(\mathcal{E}(V_{\text{pre}}, p, V_{\text{suf}}; \ \Omega_{\text{IPGO}})) - mean(\mathcal{T}(p))\|_2^2. \tag{8}$$

The $mean(\cdot)$ is defined by $mean(\{\mathbf{v}_i\}_{i=1}^L) = \frac{1}{L}\sum_{i=1}^L \mathbf{v}_i$, where $\{\mathbf{v}_i\}_{i=1}^L$, $\mathbf{v}_i \in \mathbb{R}^d$, is the input set of text embeddings. Then the optimization loss to be minimized, conditioned on $x_0$, becomes:

$$\mathcal{L}(\Omega_{\text{IPGO}}) = -\mathcal{S}\left(x_0, p\right) + \gamma P_{\text{conf}}, \tag{9}$$

where $\mathcal{S}(x_0, p)$ is one of the CLIP, HPSv2, and Aesthetics reward scores as a function of the image $x_0$ and prompt $p$, and $\gamma$ is the weight of the penalty. In our experiments we set $\gamma = 1e-3$, and increase it to $\gamma = 70$ to prevent reward hacking for the LAION aesthetics reward.

**The Outline of the IPGO Algorithm.** Here we delineate the algorithm of IPGO with all constraints and parameterization designs included.

---

**Algorithm 1** IPGO

**Input:** Raw prompt $p$, prefix/suffix generator $G_{\text{pre}}(\Omega_{\text{IPGO}})/G_{\text{suf}}(\Omega_{\text{IPGO}})$ controlled by $\Omega_{\text{IPGO}}$, text encoder $\mathcal{T}(p)$, diffusion model $x \sim q_{\text{image}}(\cdot)$, $z_T$ the initial latent noise, image reward model $S(x, p)$, conformity penalty coefficient $\gamma$, learning rate $\eta$, number of epochs $Epochs$.
**Output:** Optimal prefix/suffix generators $G_{\text{pre}}^\circ/G_{\text{suf}}^\circ$.
**for** $i = 0$ **to** $Epochs$ **do**
    Original prompt embedding: $V_0 = \mathcal{T}(p)$.
    Prefix: $V_{\text{pre}} = G_{\text{pre}}(\Omega_{\text{IPGO}})$
    Suffix: $V_{\text{suf}} = G_{\text{suf}}(\Omega_{\text{IPGO}})$
    Insert prefix and suffix: $\mathcal{E}(V_{\text{pre}}, p, V_{\text{suf}}; \ \Omega_{\text{IPGO}}) = V_{\text{pre}} \oplus V_0 \oplus V_{\text{suf}}$.
    Sample image: $x_0 \sim q_{\text{image}}(x_0|\mathcal{E}(V_{\text{pre}}, p, V_{\text{suf}}; \ \Omega_{\text{IPGO}}), z_T)$.
    Compute reward: $r = \mathcal{S}(x_0, p)$.
    Compute objective: $\mathcal{L} = -r + \gamma P_{\text{conf}}$.
    Compute gradient: $g = \nabla_{\Omega_{\text{IPGO}}}\mathcal{L}$.
    Update prefix and suffix: $\Omega_{\text{IPGO}} \leftarrow \Omega_{\text{IPGO}} - \eta g$.
    Enforce orthonormality and Value constraints.
**end for**
Return $G_{\text{pre}}^\circ(\Omega_{\text{IPGO}})$ and $G_{\text{suf}}^\circ(\Omega_{\text{IPGO}})$.

---

# G ADDITIONAL ABLATION STUDIES

In addition to the ablation experiments in the main text, we design three more ablation scenarios to investigate the effects of the lengths of prefix and suffix, size of the base text embeddings, and the relationship between the prefix/suffix lengths and the raw prompt length.

For these additional ablations we again use the Stable Diffusion v1.5 as the base diffusion model, with number of inference steps 30. The Adam optimization starts with a learning rate $1e-3$ with a decay factor 0.9 at every 10 steps.

**Prefix and suffix lengths.** Next, we test all combinations of prefix and suffix lengths of 0, 5, 10, 15, or 20 embeddings, excluding the $(0, 0)$ combination. Note that these scenarios include cases with only a prefix ($N_{suf} = 0$), or only a suffix ($N_{pre} = 0$). We conduct experiments with the full COCO dataset of 300 prompts and use the alignment CLIP score as the reward. We sample the images with 30 inference steps and optimize with 30 epochs.

Table 9 contains the results. First, by comparing the average rewards from the scenarios with the same total number of embeddings (e.g. $(5, 15)$, $(10, 10)$ and $(15, 5)$), we find that *equal prefix and suffix lengths tend to give a better performance*. Second, a longer prefix and suffix do not necessarily bring more performance gains, as illustrated by the difference between the performance of $(10, 10)$ and $(15, 15)$. Hardly any improvements occur for prefix/suffix lengths larger than 10. A very long prefix

and suffix can lead to over-parameterization which damages the image semantics and may result in reward hacking, as illustrated in the Appendix J. Therefore, in our experiments we use moderate prefix/suffix lengths of 10/10.

| $N_{suf}$ \ $N_{pre}$ | 0 | 5 | 10 | 15 | 20 |
|---|---|---|---|---|---|
| 0 | | 0.281 | 0.286 | 0.287 | 0.288 |
| 5 | 0.284 | 0.284 | 0.286 | 0.284 | 0.289 |
| 10 | 0.286 | 0.284 | 0.289 | 0.283 | 0.287 |
| 15 | 0.285 | 0.284 | 0.288 | 0.288 | 0.287 |
| 20 | 0.288 | 0.288 | 0.289 | 0.288 | 0.290 |

Table 9: Average CLIP scores for various combinations of prefix length, $N_{\text{pre}}$ and suffix length, $N_{\text{suf}}$.

**Varying $m = m_{\textbf{pre}} = m_{\textbf{suf}}$.** We first investigate the effect of $m_*$, the size of the learnable base text embeddings for the prefix and suffix. We randomly selected 30 prompts from the COCO dataset, and we conduct ablation studies with $m = 150, 300, 600$, which are about 20%, 40% and 80% of the text embedding space of dimension 768. As reward models, we choose the CLIP reward and the human preference reward (HPS). The prefix and suffix lengths are both 10.

| Reward | $m = 150$ | $m = 300$ | $m = 600$ |
|---|---|---|---|
| CLIP | 0.287 | 0.296 | 0.303 |
| HPS | 0.263 | 0.266 | 0.267 |

Table 10: Results of ablations on $m_{\text{pre}}$ and $m_{\text{suf}}$, the sizes of the sets of the base text embeddings of prefix and suffix.

Table 10 shows the results. Not surprisingly, more parameters lead to larger performance gains, shown for both rewards. However, it is interesting to see a diminishing margin of performance gain when we increase $m$. The performance improvement from increasing $m$ from 300 to 600 is less substantial than the improvement gained by increasing $m$ from 150 to 300, in particular for the HPS reward. We conclude that $m = 300$ achieves a good balance between the number of total parameters and the final performance.

**$N_{\textbf{pre}}$ and $N_{\textbf{suf}}$ based on Raw Prompt Length.** Next, we test the relationship between the length of the base prompt and the lengths of prefix and suffix. We use the CLIP reward for optimizations. We choose 30 prompts among which the first 10 prompts are simple prompts, such as "Man", "Woman" and "Student", the second 10 prompts are medium-complexity prompts for similar topics as the 10 simple prompts, selected from the COCO dataset. The last 10 prompts are even more complex versions of the second 10 prompts generated by inquiring ChatGPT with "Could you make the following 10 prompts more complex:". For example, the complex version of "A person walking in the rain while holding an umbrella" is "A middle-aged person in a long, tattered trench coat walks down a cobblestone street, their brightly colored umbrella catching the dim glow of streetlights as rain cascades around them." We make sure that the complex prompts do not exceed the limit of 77 tokens imposed by SDv1.5.

We optimize each prompt with $N_{\text{pre}} = N_{\text{suf}} = N \in \{2, 10, 15\}$ with respect to the CLIP reward. For each prompt, we record the value of $N$ that improves the output image the most. Then we calculate their proportions of each $N \in \{2, 10, 15\}$. Finally, as the evaluation metric, we use the distribution of the proportions of prompts with $N = 2, 10, 15$ as their best prefix and suffix lengths in each prompt group. We denote this distribution as $D_N(2, 10, 15)$: $D_N(2, 10, 15) = (a\%, b\%, c\%)$ means that $a\%$ (or $b\%$ or $c\%$) of the prompts in the target prompt group have $N = 2$ (or $N = 10$ or $N = 15$) as their best prefix/suffix lengths.

From the ablation results, *we do not observe any correlation between the prompt length and the prefix and suffix lengths.* The simple prompt group has $D_N(2, 10, 15) = (30\%, 50\%, 20\%)$; the medium-complex prompt group has $D_N(2, 10, 15) = (50\%, 20\%, 30\%)$; and the complex prompt group has $D_N(2, 10, 15) = (20\%, 40\%, 40\%)$. For robustness, we test IPGO on a very long prompt

in Appendix H. Generally, we recommend using fewer inserted embeddings for very short prompts to avoid over-parameterization (an illustration of how it may lead to reward hacking is in Appendix J).

## H    IPGO FOR A VERY LONG PROMPT

We illustrate IPGO on a very long prompt to see whether the optimized prefix/suffix may crowd out some of the many details in these long prompts. We use the following pair of prompts, one is a short version and the other is the long version (over 150 words), as input into FLUX 1.0-Schnell. The two prompts are the following:

> **Short:** *A fluffy baby sloth having a tiny tea party on a mushroom in an enchanted forest, surrounded by glowing wildflowers. Soft, warm lighting, hyperrealistic, adorable.*

> **Long:** *Generate a high-resolution, ultra-detailed photograph of a tiny, fluffy baby sloth enjoying a miniature tea party in a fantastical enchanted forest. The sloth, no bigger than a teacup, should have exceptionally soft, light brown fur with subtle, almost invisible, streaks of pale green moss clinging gently to its back, hinting at its natural habitat. Its eyes should be large, glistening with childlike wonder and a hint of sleepy contentment, framed by delicate, almost translucent, white fur. The tea party setup is equally magical: imagine a small, hand-carved mushroom acting as the table, adorned with a tiny, intricately embroidered lace doily. On the table, there's a delicate porcelain teacup, no larger than a thimble, filled with what appears to be a shimmering, honey-colored liquid. Beside it, a single, minuscule shortbread cookie shaped like a star, perfectly golden brown. The forest floor around the tea party should be a carpet of vibrant, dewy moss, interspersed with glowing bioluminescent wildflowers in shades of lavender and soft blue.*

To illustrate, we optimize the image based on the Human Preference Score. We inserted the 20 prefix and 20 suffix tokens in the T5-v1.1-XXL text embeddings, each with projection dimension 150 compared with the 4096 dimensional text embedding space.

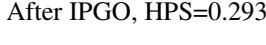

| Before IPGO, HPS=0.288 | After IPGO, HPS=0.293 |

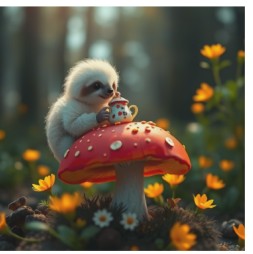 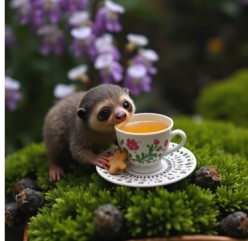

Table 11: Images produced by FLUX 1.0-Schnell, without (left) and with (right) IPGO prompt optimization for human preference.

The generated images are shown in Table 11. IPGO improves the Human Preference Score by 1.7%, from 0.288 to 0.293. Note that after IPGO optimization, the image not only has a higher preference score, but also more accurately aligns with attributes such as brown fur, the size of the sloth, the lavender, the carpet of moss, the porcelain cup, the honey-colored liquid, and the cookie with a star-shape, as specified in the long prompt. The mushroom table is not captured, however, and image alignment can be further improved by increasing the weight on the conformity penalty. We will further investigate this in future research.

## I    REWARD HACKING EXAMPLES

In table 12 we provide two examples of prompt-image pairs for which we found evidence of reward hacking because of over-optimization of the aesthetics reward. We show the images produced with a small and a large weight of the conformity penalty, which illustrates that that penalty helps mitigates reward hacking.

**Prompt:** "A man with a bike at a marina"

$\gamma = 0.001$ $\gamma = 70$

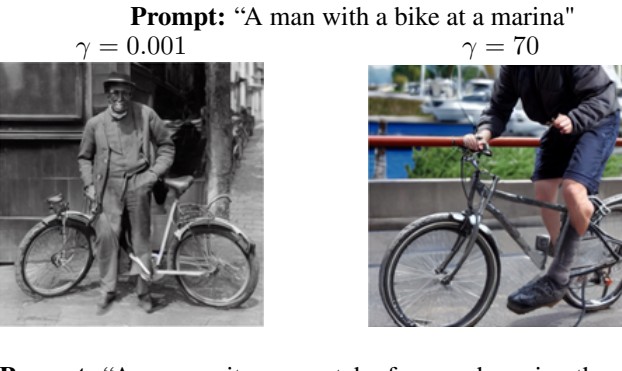

**Prompt:** "A person sits on a patch of grass observing the scenery"

$\gamma = 0.001$ $\gamma = 70$

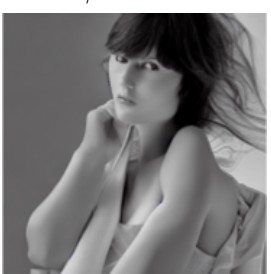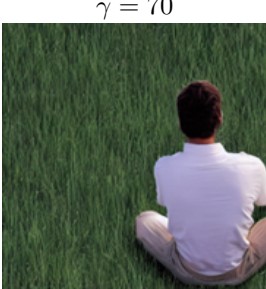

Table 12: Prompt-image pairs with evidence of reward hacking for the aesthetics reward with a small weight ($\gamma = 0.001$) of the conformity penalty; a large weight ($\gamma = 70$) mitigates the hacking problem.

## J  OVERPARAMETERIZATION AND REWARD HACKING

When the lengths of the prefix and suffix significantly exceed that of the base prompt, IPGO faces the risk of over-parameterization, which may exacerbate the problem of reward hacking. Table 13 illustrates this issue, showing the evolution of images generated during the optimization for aesthetics improvement of the simple base prompt "cat", which only has one single token, with a very long prefix and suffix of $N_{pre} = N_{suf} = 30$. The effects of over-parametrization may be severe in the case of the aesthetics reward, for which we revealed that reward hacking is a potentially serious problem, see Tables 4 and 12. In the first several steps, IPGO produces images that adhere to the prompt and display a cat, but at later steps, the object in the image changes to a person, which even changes to a different person in later steps. Apparently, if the prefix and suffix are long, and for a low weight on the conformity penalty, optimization of the inserted embeddings overwhelms the semantic structure of base prompt and harms alignment of the image with the base prompt. We recommend using shorter prefix and suffix lengths for shorter prompts, especially when reward hacking is a potential problem. Additional solutions are to increase the weight of the conformity penalty or blend the prefix/suffix embeddings of CLIP and aesthetics rewards to induce prompt-image alignment, as described in the main text.

Step 0 Step 4 Step 13 Step 32

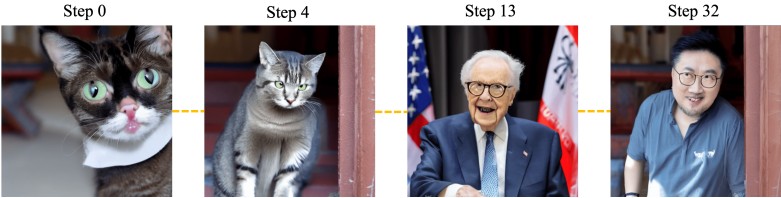

Table 13: Images generated during IPGO's optimization on the **base prompt:** "Cat", with $N_{pre} = N_{suf} = 30$ for aesthetics. Because of over-parameterization the images that are produced show evidence of reward hacking, that is, poor alignment with the *base* prompt.

# K    ADDITIONAL IMAGE EXAMPLES

All optimized images shown in this section were optimized with respect to the human preference reward HPSv2.

## K.1    ADDITIONAL IPGO COMPARISONS WITH DRAFT-1 AND TEXTCRAFTOR

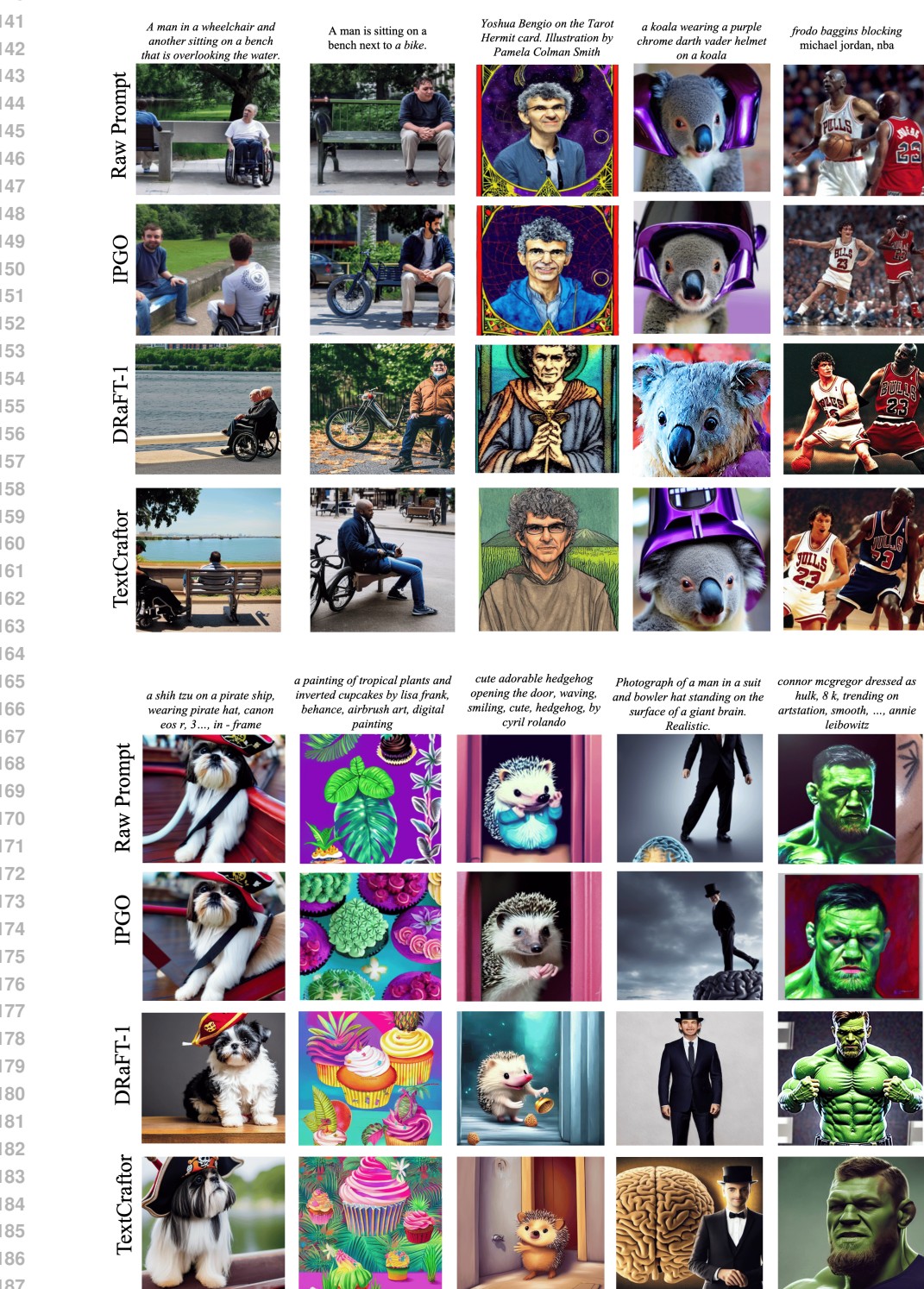

## K.2 Others

movie still from the fifth element, body portrait of a young woman jessica alba cyborg …

colorful graffiti, shards, illustration, highly detailed, simple, no jagged lines, smooth, artstation, centered artwork by shepard fairey of centered portrait of an elven

a highly detailed beautiful portrait of hamster playing poker, by gregory manchess, james gurney, james jean

These 3D portraits are unbelievably realistic. unreal engine 5 RTX raytracing nvidia hairworks render of portrait of the most beautiful girl with blue eyes.

movie film still of Alexandra Daddario as a female Colossus in a new X-Men movie, cinematic

caricature angry old man in chair inside a dark house, painting by by ralph grady james, jean christian biville

Raw Prompt
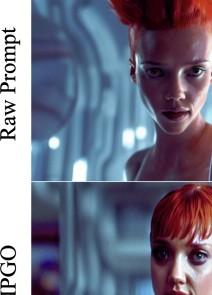 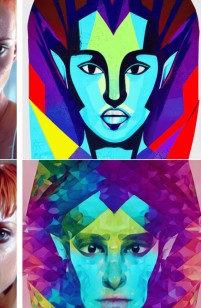 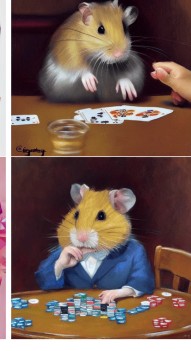 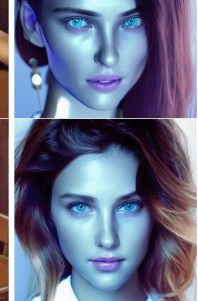 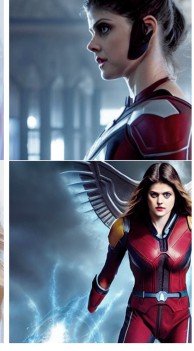 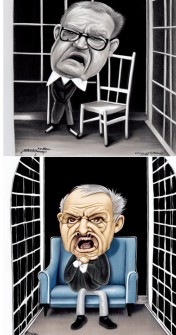

IPGO

photo of a group of female doctors, working in a hospital

a pov shot, color cinema film still of saul goodman & katy perry in blade runner 2 0 4 9, cinematic lighting at night.

70 mm portrait, furry rocket the raccoon sitting in the cockpit of the millennium falcon, … photorealism!!

futuristic utopian paradise, canals, bridges, white marble temples, palm trees, … cinematic lighting,, pinterest

a shinto shrine path atop a mountain,spring,cherry trees,beautiful,nature,distant shot,random angle

backlit levitating geert wilders raising both his arms amid a crowd, aesthetic

Raw Prompt
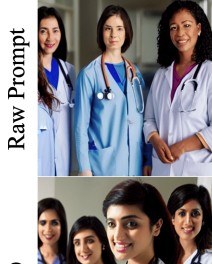 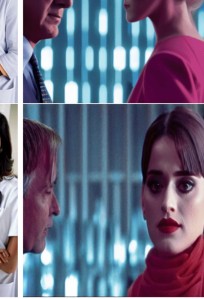 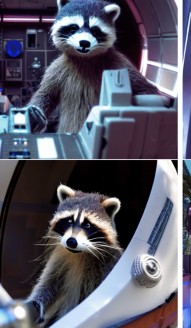 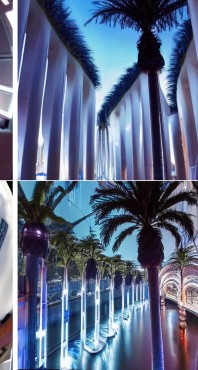 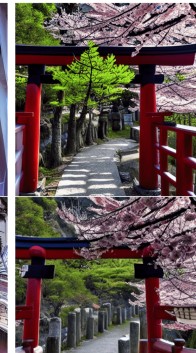 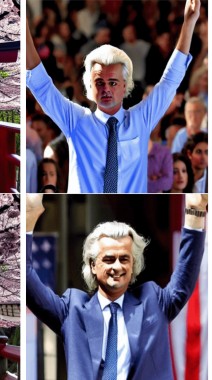

IPGO

a highly detailed symmetrical painting of a female sorcerer with piercing eyes in a dungeon, … glenn fabry

RAW photo of a cute cat as a cowboy standing in a desert, bokeh

Photo of a blonde 18yo cybord girl, intricate white cyberpunk respirator and armor

genere una imagen de un perro pequeño feliz , jugando y corriendo por el parque

a drawing of a girl with bright blue hair wearing sunglasses, cyberpunk art …, pop art

portrait of a Young woman with short blonde hair wearing glasses and freckles around her nose

Raw Prompt
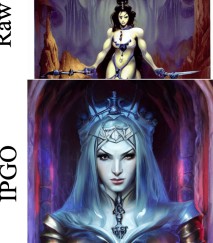 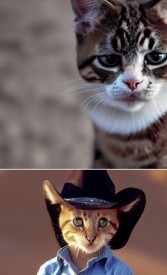 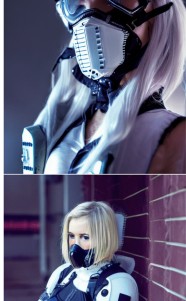 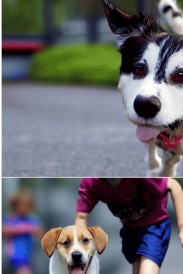 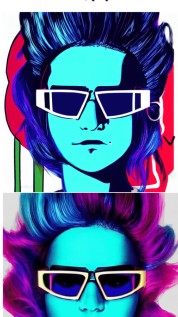 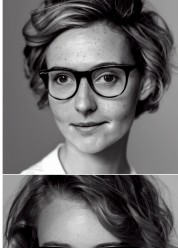

IPGO

### K.3 MORE VISUAL COMPARISONS

For each image pair, the top image is generated by SDv1.5, the bottom image is optimized by IPGO.

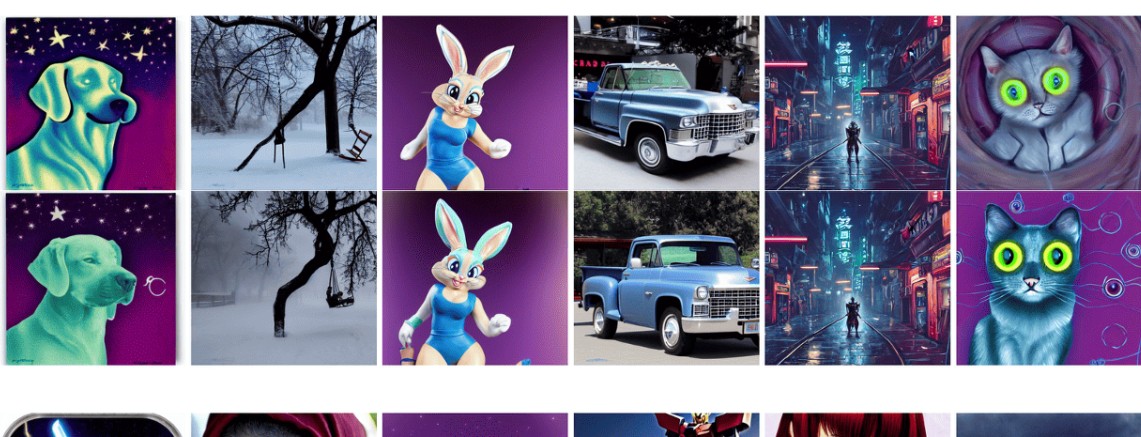

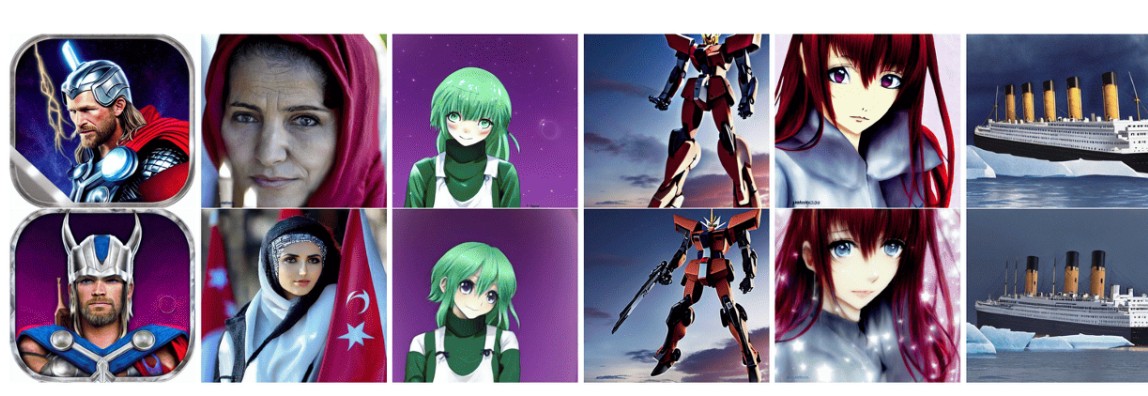

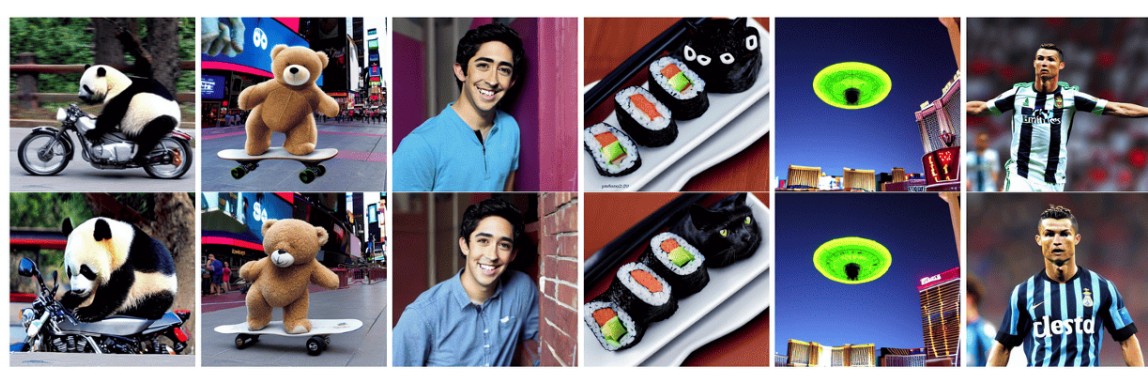

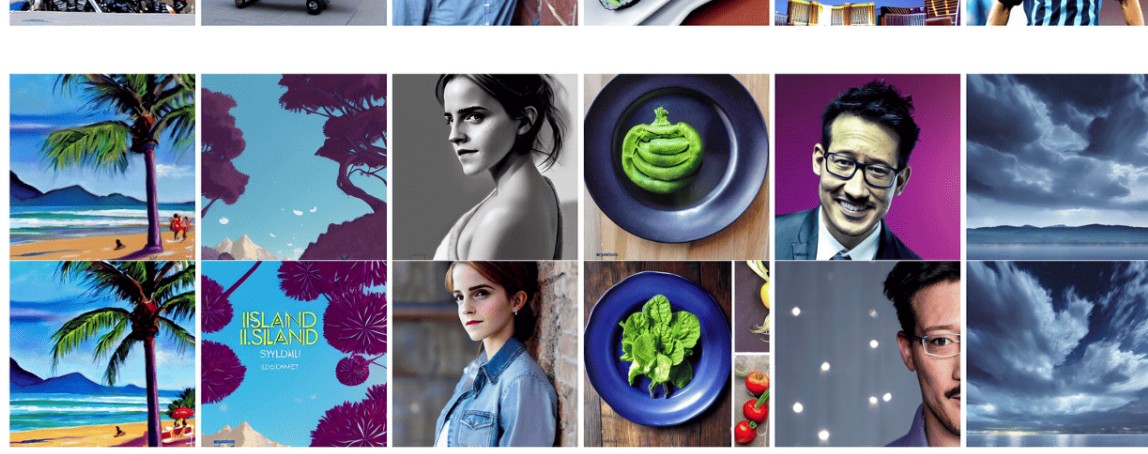

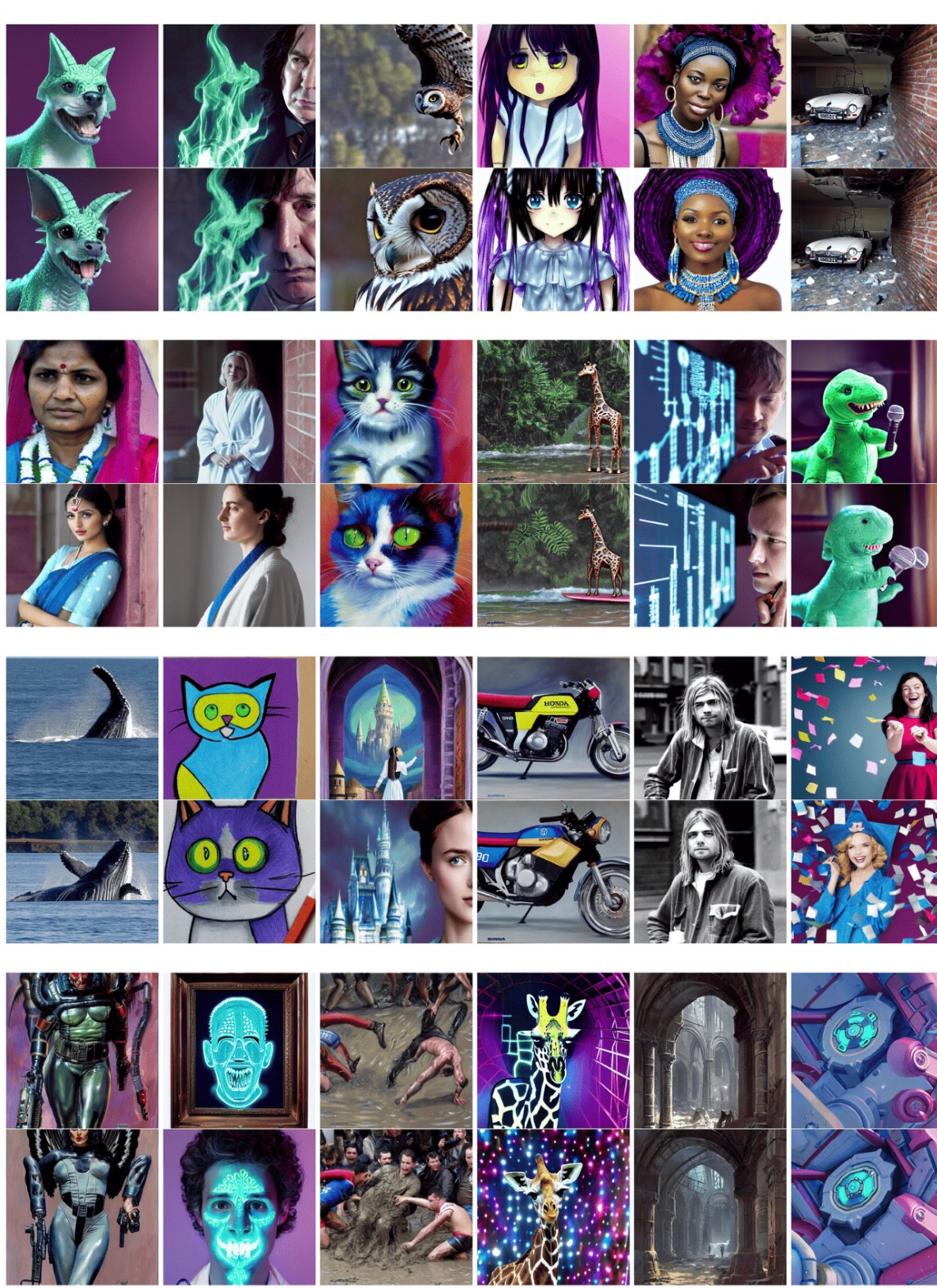

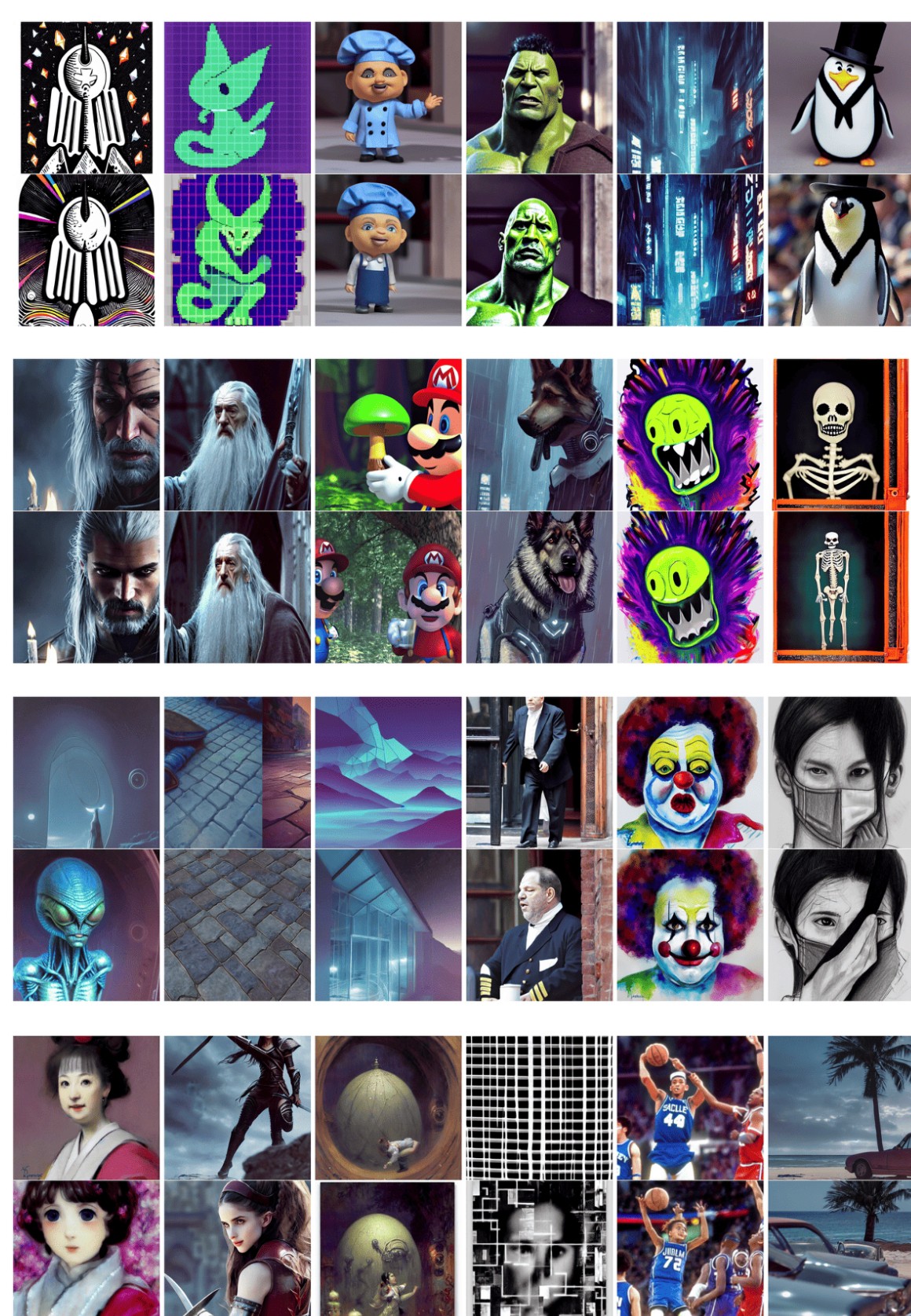

