# OpenReview forum: "IPGO: Indirect Prompt Gradient Optimization for Text-to-Image Model Prompt Finetuning"
_ICLR.cc/2026/Conference — Submitted to ICLR 2026_

### Official Review · Reviewer_fnYW · 2025-10-26

**Soundness:** 2
**Presentation:** 2
**Contribution:** 2
**Rating:** 4
**Confidence:** 4

**Summary:**

This paper introduces a prompt tunning methods for human or aesthetics preference alignment. The main idea of this paper is to finetun the prefix and suffix of the given prompt in the embedding space. Therefore, the trainable parameter can be greatly reduced. In addition, the authors introduced low-rank approximation and rotation transform on the trainable embedding space, which also demonstrate its improvement in the experiment section.

**Strengths:**

- Strengths

1) Similar to the prefix prompt tunning, this paper finetung prefix and suffix embedding for aligning the human preference using reward model.

2) In addition, the authors introduced rotation and low-rank approximation (I would like to say that it is not true) for improving the expressiveness of the trainable embedding.

**Weaknesses:**

- Weaknesses

1) In the related works, the authors claimed that they do not require to access the original image compared to PEZ and Textual Inversion. However, I think this is not true as it is related to the task. PEZ requires original images since they consider the prompt discovery tasks.

2) In equation (3), the dimension of Z and E are not aligned, they can not be multiplied directly.

3) The reason for decompose the prefix or suffix embedding into low-rank matrices are not clear. In addition, according to the Table 8, m is set to 300 while the prefix or suffix is 10, thus it is totally not the low-rank approximation or parameter-efficient strategy, but the over-complete dictionary learning. The authors should thoroughly correct this claim.

4) Given a reward loss function, how to optimize the embedding? Can we directly optimize the embedding through via BP? More details can also be introduced in the paper.

**Questions:**

See the weakness section for details.

---

> ### Author Response · Authors · 2025-11-22
>
> We sincerely appreciate your insightful comments! We include new explanations, numerical results and figures in the revise paper (labeled in each reply).
>
> > **Weakness 1:** In the related works, the authors claimed that they do not require to access the original image compared to PEZ and Textual Inversion. However, I think this is not true as it is related to the task. PEZ requires original images since they consider the prompt discovery tasks.
>
> Thank you for your detailed comments. We apologize for misrepresenting PEZ, we have corrected our explanation on **page 3** of the updated manuscript.
>
> > **Weakness 2:** In equation (3), the dimension of Z and E are not aligned, they can not be multiplied directly.
>
> Thank you for pointing out the error, Z should be $m* \times N$. We have corrected on **page 5** of the updated manuscript.
>
> > **Weakness 3:** The reason for decompose the prefix or suffix embedding into low-rank matrices are not clear. In addition, according to the Table 8, m is set to 300 while the prefix or suffix is 10, thus it is totally not the low-rank approximation or parameter-efficient strategy, but the over-complete dictionary learning. The authors should thoroughly correct this claim.
>
> We use a reduced rank representation because of parameter efficiency. We apologize and agree that “low rank” is not the correct terminology. We now refer to “parameter efficient” or “reduced rank” rather than low rank. Please note that the dimension of the original embedding, $d=756$ is much larger than $m=300$, the size of the basis text embeddings on which each prefix and suffix was constructed, so that there should not be over-complete dictionary learning. We have mentioned these dimensions in **paragraph Constrained Prefix-Suffix Tuning** on **page 4**.
>
> > **Weakness 4:** Given a reward loss function, how to optimize the embedding? Can we directly optimize the embedding through via BP? More details can also be introduced in the paper.
>
> Yes, we can directly optimize the embedding of the prefix and suffix, without constraints. Our ablation experiments show that this doesn’t work as well as the constrained optimization, see **Table 5 (Naïve learning)**. The constraints and penalty mitigate possible reward hacking.

---

### Official Review · Reviewer_FprH · 2025-11-01

**Soundness:** 3
**Presentation:** 4
**Contribution:** 3
**Rating:** 6
**Confidence:** 2

**Summary:**

The paper introduce Indirect Prompt Gradient Optimization (IPGO), a parameter-efficient method for aligning text-to-image (T2I) diffusion models with various reward objectives including aesthetic, CLIP alignment, and human preference. Instead of modifying diffusion model’s backbone or text-encoder parameters, IPGO optimizes a few learnable prefix and suffix embeddings added to the original prompt’s text embeddings. These embeddings are optimized via a constrained, gradient-based procedure incorporating low-rank approximation, rotation parameterization. The framework is training-efficient (0.47M parameters) and evaluated under multiple datasets and showcased consistent improvements of 1–3% over strong baselines.

**Strengths:**

1. **Clear Writing:** The paper is clearly written and well-structured. The motivation and methodology are presented in an organized and accessible manner, making it easy to follow and pleasant to read.
2. **Novel and Effective Method:** The proposed approach introduces a novel optimization-based framework for prompt refinement, which effectively enhances text-to-image alignment and image quality. Extensive experiments showcased the effectiveness of the IGPO method.

**Weaknesses:**

1. **Limited Exploration on Large-Scale Models:** While the method targets parameter-efficient learning, most results are on backbones with relatively small text encoders (e.g., SD). It remains unclear how well the approach scales to larger systems and richer text stacks (Flux). Evaluating on modern large diffusion models—adding both quality gains and compute/latency/memory trade-offs to Table 6—would strengthen the claim of broad effectiveness and practical scalability.
2. **Weak Justification for Using Both Prefix and Suffix:** The paper claims that using both prefix and suffix embeddings improves performance. However, as shown in Table 5, configurations with only prefix or only suffix embeddings (e.g., (10, 0) or (0, 10)) achieve nearly identical CLIP scores (0.286 vs. 0.289). This marginal difference does not strongly justify the necessity of employing both prefix and suffix components simultaneously. Authors may need to compare (10,10) with (20,0) or (0,20) to ensure a fair comparison with the same learnable parameters.

**Questions:**

see weakness

---

> ### Author Response · Authors · 2025-11-22
>
> We sincerely appreciate your constructive comments! We include new explanations, numerical results and figures in the revise paper (labeled in each reply).
>
> > **Weakness 1:** Limited Exploration on Large-Scale Models
>
> Thank you for your positive comments. IPGO is applicable to a wide range of models. We used SD1.5 for computational efficiency, and because it has also been used to evaluate other recent methods (see for example Zhai et al. 2025). We now position IPGO as a model-independent method that can be applied in conjunction with several diffusion models. We showed in **section 6.3** and **Table 6** that IPGO can also be applied to SD3,  SDXL, and now also add results for FLUX-Schnell. IPGO can thus be applied to any prompt-conditional diffusion model that has a text encoder component.
>
> The optimization time for IPGO, under the current setting, is **3-5 minutes** for each single-prompt optimization (with L4 GPU), for 30 optimization steps in total. The vast majority of the clock time is taken up by the image generation steps. Since in our single-prompt optimization scenario, all our gradient-based baselines were trained with the same #optimization steps/#image generations, we focus on the parameter-efficiency of the IPGO framework. We demonstrate that in **Figure 5 in Appendix C**.
>
>
>
> > **Weakness 2:** Weak Justification for Using Both Prefix and Suffix
>
> We have added comparisons with (10,10), (20,0), (0,20) and (20,20) settings to **Table 9 of Appendix E** in the revised manuscript. We found (20,20) delivers a better performance 0.290 compared to 0.289 of (10,10), however the difference is small and there is a notable increase in the number of trainable parameters.
>
> Although the gain is relatively small, the advantage of having balanced prefix and suffix is consistently shown in our results. Therefore, we conclude that having a prefix and suffix of the same length is a recommended practice for IPGO optimization.
>
> In order to prevent over-parametrization and reward hacking (which we now address more extensively in the paper) we recommend using more moderate prefix and suffix lengths of (10,10).

---

### Official Review · Reviewer_aZrb · 2025-11-01

**Soundness:** 3
**Presentation:** 3
**Contribution:** 2
**Rating:** 4
**Confidence:** 4

**Summary:**

This paper introduces Indirect Prompt Gradient Optimization (IPGO), a parameter-efficient framework for prompt-level finetuning in text-to-image (T2I) diffusion models. IPGO enhances prompt embeddings by injecting learnable prefix and suffix embeddings, optimized via gradient-based methods with low-rank approximation, rotation, and stability constraints (orthonormality, range, and conformity). Unlike prior approaches, IPGO requires no modification to the diffusion model or text encoder and operates with far fewer parameters, enabling efficient, prompt-wise optimization at inference. Experiments across three datasets and three reward models (aesthetics, image-text alignment, and human preference) show that IPGO consistently outperforms baselines, such as TextCraftor and DRaFT-1, while using fewer parameters.

**Strengths:**

+ The proposed method is efficient and it achieves better results with fewer parameters and lower hardware requirements than baselines such as TextCraftor and DRaFT-1.

+ Looks like the proposed method has potential to be applied to existing T2I diffusion models and reward functions without modifying the underlying model or text encoder.

+ The proposed outperforms several baselines across multiple datasets and reward types. It also provides ablation studies to validate the contribution of each design choice and constraint.

**Weaknesses:**

- Native image generation models (e.g. VAR [a] BAGEL [a], ) are pretty popular recently which does not include a text encoder. It is not clear how the proposed method can be applied or generalized to these recent SOTA methods.

[a] Visual Autoregressive Modeling: Scalable Image Generation via Next-Scale Prediction, 2024
[b] Emerging properties in unified multimodal pretraining, 2025

- It is not clear how well the proposed method can handle the case where inference prompts are long and detailed (e.g. > 150 words). As for recent methods, LLM prompt rewrite/expansion is a commonly used strategy which will transfer short input prompt to detailed long prompt as input to the model. It is not clear how good ADSS can be applied to those cases with detailed prompt.

- SOTA performances are claimed by the submission, while the most recent methods included for comparisons are TextCraftor (year 2024) and DRaFT-1 (year 2023) which is a bit old.

**Questions:**

Please refer to the detailed questions raised in Weakness section above.

---

> ### Author Response · Authors · 2025-11-22
>
> We greatly appreciate your helpful comments! We include new explanations, numerical results and figures in the revise paper (labeled in each reply).
>
> > **Weakness 1:** Native image generation models (e.g. VAR [a] BAGEL [a], ) are pretty popular recently which does not include a text encoder. It is not clear how the proposed method can be applied or generalized to these recent SOTA methods.
>
> Because it does not allow for prompts, IPGO cannot be used with VAR. It is applicable to BAGEL in principle, but the application is technically nontrivial, because BAGEL is not a dedicated T2I model.  We now show in **Table 6** of **section 6.3** that IPGO can be used with FLUX-Schnell, another recent, very strong SOTA model.
>
> > **Weakness 2:** It is not clear how well the proposed method can handle the case where inference prompts are long and detailed (e.g. > 150 words).
>
> IPGO can be applied to very long prompts. In **Appendix H** of the revised manuscript, we tested our IPGO method for a prompt with >150 words on FLUX 1.0-Schnell model and showed numerical and visual results. We used Human Preference Score as a reward. FLUX models take two prompts for each image generation: one short prompt and the other the long prompt (164 words). We find that IPGO improves the reward from 0.288 to 0.293, and the generated images are visually more aligned with the prompt.
>
> > **Weakness 3:** SOTA performances are claimed by the submission, while the most recent methods included for comparisons are TextCraftor (year 2024) and DRaFT-1 (year 2023) which is a bit old.
>
> To our best knowledge, TextCraftor is still the most recent SOTA that has a focus on the text embedding space. We’re happy to try others if you have specific suggestions.

---

### Official Review · Reviewer_gMyg · 2025-11-04

**Soundness:** 1
**Presentation:** 2
**Contribution:** 3
**Rating:** 2
**Confidence:** 4

**Summary:**

This paper proposes IPGO (Indirect Prompt Gradient Optimization), a method for aligning text-to-image diffusion models with reward functions through optimization of prompt embeddings. The approach adds learnable prefix and suffix embeddings to the original prompt embeddings, parameterized through rotated low-rank approximations. Instead of using explicit KL regularization like traditional RL-based alignment methods, IPGO relies on three embedding-space constraints to prevent reward hacking: orthonormality of the embedding basis, range constraints on coefficients ([-1,1]), and conformity (mean preservation with original prompt embeddings). The method operates as a test-time optimization approach, optimizing each individual prompt separately for multiple epochs. Experiments are conducted on prompts from COCO, DiffusionDB, and Pick-a-Pic datasets using Stable Diffusion v1.5 and three reward models (CLIP alignment, LAION aesthetics, HPSv2 human preference). IPGO is compared against six baselines, including training-based methods (TextCraftor, DRaFT-1, DDPO) and training-free methods (DPO-Diff, Promptist), showing improvements over competing methods when evaluating on the reward that is optimized.

**Strengths:**

- **Novel parameterization approach for prompt embedding optimization**: The method combines prefix-suffix embeddings with rotated low-rank parameterization and three constraints (orthonormality, range, conformity) to keep optimization within a meaningful embedding region. The linguistic motivation for prefix-suffix design is intuitive, and the approach allows preserving original prompt semantics while adding learnable content.

- **Ablation studies**  The paper provides valuable ablations showing task-dependent importance of constraints (e.g., range crucial for aesthetics, orthonormality for alignment) and demonstrating that the parameterization significantly outperforms naive unconstrained optimization. Ablations on prefix/suffix lengths reveal that equal lengths work best and longer isn't necessarily better.

**Weaknesses:**

Major weaknesses, experimental design: (I would be open to reconsidering my score given that these concerns are adressed.)
- **No evaluation on general T2I benchmarks**: The evaluation is limited to the same reward models used for optimization, which provides no evidence that IPGO improves general image quality beyond simply overfitting to the metric. Without testing on standard T2I benchmarks that assess compositionality and attribute binding (e.g., GenEval, T2I-CompBench), or conducting human studies, or minimally cross-validating rewards,  the reported gains could reflect reward hacking rather than genuine improvements. The risk of exploiting reward model biases is substantial, especially given the multi-epoch per-prompt optimization and the absence of explicit KL regularization to maintain fidelity to the original model distribution.
- **Experimental design does not align with the test-time optimization paradigm**: The experimental design is not fully convincing to me, IPGO requires multiple optimization epochs per generated image, against training-based baselines (e.g., TextCraftor, DRaFT-1) that offer instant inference after a one-time training cost. While this comparison is interesting, I would argue the main comparison should be against other test-time techniques, including wall-clock time as one axis. Currently, it's unclear what the compute vs performance trade-off looks like. While the main comparison would be Promptist, other test-time optimization techniques are in my view the main competing methods to IPGO (e.g., noise selection [1] (Best-of-N, or over paths) or noise optimization [2,3]).
- In my opinion, reporting wall-clock time (and GPU memory) and performance on some metric that is disentangled from the optimized metric (e.g. GenEval) compared to more test-time techinques, is needed to accurately assess the performance of IPGO, which the current evaluation doesn't sufficiently cover.

Minor weaknesses:
- **Limited justification for specific design choices**: Many of the method's core components lack rigorous justification and appear arbitrary. The rotation parameterization is motivated by a simplified 2D analysis that does not transfer to the high-dimensional setting and is empirically contradicted by results where it harms alignment scores. Likewise, the constraint formulations, such as the `[-1,1]` range and the preservation of the mean, are presented as heuristics without theoretical motivation or ablation against alternatives. While the paper shows these components contribute to performance, it fails to provide a clear rationale for why these specific choices are optimal.
- The majority of experimental evaluations are only on SD1.5, which is far w.r.t. performance from the models currently used in practice (SD3, FLUX, ...).

[1] Ma et al. "Inference-Time Scaling for Diffusion Models beyond Scaling Denoising Steps". CVPR 2025.

[2] Tang et al. "Inference-Time Alignment of Diffusion Models with Direct Noise Optimization". ICML 2025

[3] Eyring et al. "ReNO: Enhancing One-step Text-to-Image Models through Reward-based Noise Optimization". NeurIPS 2024.

**Questions:**

- **Generalization across noises**: Does an optimized prefix/suffix for a given prompt generalize to different initial noise seeds, or must the optimization be re-run for every new image generation? What is the typical wall-clock time for this inference-time optimization?
- **Interpretability of Embeddings**: Have you attempted to project the learned prefix/suffix embeddings back into the vocabulary space to see if they correspond to interpretable words or concepts? What does IPGO learn to "say" to improve aesthetics or human preference?
- **Rotation's Role**: The rotation component appears to have a task-dependent effect (improving aesthetics but not alignment in the ablation). Do you have an intuition for why this is the case? Could the rotation be made adaptive based on the reward function?

---

> ### Author Response · Authors · 2025-11-22
>
> Thank you very much for the detailed comments! We apologize for extending our reply to multiple comments and any inconvenience this brought. We try to be as concise as possible.
>
> Our new explanations, numerical results, and figures are in the revised manuscript (labeled for each reply).
>
> > **Weakness 1:** No evaluation on general T2I benchmarks
>
> We sincerely appreciate your crucial comment on the potential reward hacking problem. We acknowledge that problem in the paper and cite the relevant literature. We address in the following ways **(page 8, Table 4, Appendix F)**:
> 1. **The Range and Conformity constraints are acting to mitigate reward hacking**: The constraints of the IPGO framework are formulated to minimize the semantic distortion by the prefix and suffix that causes reward hacking:
>     - (a) *Range Constraint* confines the optimization space on each linear coefficient $Z_i$ (equation 3) to be within [-1,1] to restrict text embedding length. Our experiment with range [-8,8] (because mean length of CLIP text embeddings=16) shows CLIP drops to <0.20 on COCO. (**page 4**)
>     - (b) *Conformity Penalty* penalizes drift from the base prompt’s mean embedding, reducing semantic drift and reward hacking. This conformity penalty comes from the observation that the mean text embeddings reflect the semantics of the base prompt. t-SNE projections of mean text embeddings in **Fig. 4** illustrate this effect.
>
> 2. **Cross Validations on the aesthetics and human preference optimization with CLIP and VQA scores**: We thank you again for suggesting cross validations to evaluate the potential reward hacking issue in our results. We used VQAScore[1], another strong benchmark for image quality evaluation, to detect reward hacking (**Table 4**). We are sorry but we had difficulty in implementing GenEval and T2I-CompBench you suggested. We found:
>    - **For CLIP / HPS:** We found no evidence of reward hacking, since VQAScores improve after IPGO optimization for both cases.
>    - **Aesthetics:** Reward hacking occurs due to a drop in VQAScore. This problem arises due to the very small weight on the conformity penalty (γ=0.001) which in order for a fair comparison to other baselines in **Table 2**. **Table 2** demonstrates the maximum rewards of an almost penalty-free reward optimization, which however may lead to reward hacking. We propose the following to address this problem. :
>        - **In-training penalty:** We increased conformity penalty from 0.001 → 70, improving VQAScore (0.575 → 0.727) on COCO. We also illustrate the visual effect of the penalty in **Figure 6 in Appendix G**. Such large penalty is necessary since aesthetics does not have semantic info.
>        - **Post-training blending:** We propose a second method of regularization post-training, which does NOT require retraining the model. IPGO allows the blending of the trained prefix and suffix. We blended Aesthetics/CLIP prefixes–suffixes (weights 0.8/0.2), then followed by Best-of-N (N=5). This method restores VQAScore while improving aesthetics (0.505 → 0.544 on COCO).
>
> Table 4 in the revised manuscript:
> | Dataset | SD v1.5 | IPGO-CLIP | IPGO-HPSv2 | IPGO-LAION | IPGO-Blend |
> |:---|:---|:---|:---|:---|:---|
> | **COCO** | 0.741 | 0.790 | 0.773 | 0.575 | 0.786 |
> | **DiffusionDB** | 0.716 | 0.761 | 0.722 | 0.615 | 0.745 |
> | **Pick-a-Pic** | 0.650 | 0.714 | 0.689 | 0.594 | 0.702 |
>
> > **Weakness 2:** Experimental design does not align with the test-time optimization paradigm
>
> Thank you for pointing out this weakness. We would like to clarify why we design our experiments in this way.
>
> 1.**GPU Constraint and Fair Comparisons:** Colab GPU limits require single-prompt optimization, which avoids generalization cost and enables fair comparison with reward-gradient baselines (DRaFT, TextCraftor, DDPO). We also include Promptist and DPO-Diff. **Section 5.2** now explains this more clearly, and we corrected the terminology to “single-prompt optimization.”.
> 2.**Focus on the text embedding space:** To isolate effects of prompt-text embeddings, we fix the noise seed; noise variation has major impact. Noise-optimization methods (e.g., Best-of-N) therefore lie outside our scope. We now explicitly cite relevant work.
>
> *Nonetheless*, we compared IPGO (prefix/suffix transferred across seeds) with Best-of-N (N=30):
> - **CLIP:** 0.295 (BoN) vs 0.301 (IPGO)
> - **HPS:** 0.289 (BoN) vs 0.289 (IPGO)
> This suggests IPGO remains competitive even without leveraging noise diversity.
>
> > reporting wall-clock time (and GPU memory) and performance on some metric
>
>  Single-prompt IPGO takes **3–5 minutes** on an L4 GPU for 30 steps; most time is on image generation. Since all baselines use identical step/image budgets, we emphasize IPGO’s parameter efficiency, as illustrated in the **Figure 5 (Appendix C)**.
>
> [1] Lin, Zhiqiu, et al. "Evaluating text-to-visual generation with image-to-text generation." European Conference on Computer Vision. Cham: Springer Nature Switzerland, 2024.

---

> > ### Author Response · Authors · 2025-11-22
> >
> > > **Minor Weakness 1:** Limited justification for specific design choices
> >
> > Please also see the response for your first comment for our motivation for the range constraint and conformity penalty. We have provided a stronger rationale for these in the paper along the following lines:
> >
> > 1.	Orthogonality is imposed to ensure each dimension represents a semantic concept that is disentangled from others and to prevent semantic leakage.
> > 2.	Rotation is applied to align the semantic meaning in the prefix and suffix embeddings with the original prompt while preserving the embedding vector angles and norms; note that our formulation is motivated by the fact that any orthogonal rotation matrix can be decomposed in a sequence of pairwise rotations[2]. Rotation also leads to more stable training.
> > 3.	Range constraints improve training stability, prevent exploding gradients, and are a regularization that restricts the optimization, mitigating reward hacking.
> > 4.	Conformity penalty act as a regularization (similar to KL) to promote coherence of the embeddings of the prefix and suffix with the original prompt, thus also mitigating reward hacking (**page 5, paragraph conformity constraint**)
> >
> > [2] Givens, Wallace. "Computation of plain unitary rotations transforming a general matrix to triangular form." Journal of the Society for Industrial and Applied Mathematics 6.1 (1958): 26-50.
> >
> > > **Minor Weakness 2:** The majority of experimental evaluations are only on SD1.5, which is far w.r.t. performance from the models currently used in practice (SD3, FLUX, ...)
> >
> > IPGO is applicable to a wide range of models, we used SD1.5 for computational efficiency, and because it has also been used to evaluate other recent methods (see for example Zhai et al. 2025).
> >
> > We now position IPGO as a model-independent method that can be applied in conjunction with several diffusion models. We showed in **section 6.3** that IPGO can also be applied to SD3,  SDXL, and now also add results for FLUX-Schnell to **Table 6**. IPGO can thus be applied to any prompt-conditional diffusion model that has a text encoder component.
> >
> > > **Question 1:** Generalization across noises
> >
> > We now demonstrate efficient transfer to different noise seeds (see also our reply above). Thank you for this suggestion. We also report parameter efficiency and compute times (see our reply above).
> >
> > > **Question 2:** Interpretability of Embeddings
> >
> > We have made some attempts to map the prefix/suffix embeddings back to the vocabulary space but have not been successful. Interpretation of the prefix/suffix embeddings is one of our future research directions.
> >
> > > **Question 3:** Rotation's Role
> >
> > Rotation aligns the semantic meaning in the optimized prefix and suffix embeddings with the original prompt while preserving the embedding vector angles and norms. The conformity penalty already induces alignment **(see Figure 2)**, which may make the rotation redundant for the alignment reward.
> >
> > In addition, the rotation could potentially make the optimization more efficient since it can keep tracing the shortest path from the initial point to the optimal point in 2D spaces **(see Appendix A)**. Rotation could be made adaptive targeting different reward functions, but we have not pursued this.

---

### Author Response · Authors · 2025-11-22

Dear Reviewers,

We sincerely thank you for your thoughtful and constructive feedback! We are very sorry to reach out to you on weekend. We have dedicated considerable time to crafting this rebuttal, aiming to clarify our framework designs and thoroughly expand both our experimental results on various areas, which significantly enhanced the paper. The specific areas addressed include:

- Reward Hacking (Reviewer gMyg)
- Comparisons with Best-of-N (Reviewer gMyg)
- Design Choices (Reviewers gMyg, FprH and fnYW)
- Generalizations to larger-scale models (Reviewers gMyg, aZrb and FprH)

We have revised our paper with the more explainations, additional experimental results and figures in the revised PDF.

Once again, we are grateful for your valuable insights, which have significantly enhanced our work. We hope our replies comprehensively address all concerns.

Thank you for your time and consideration.

Sincerely,

The Authors

---

### Author Response · Authors · 2025-12-03
**Summary of Contributions and Responses**

Dear AC,

We sincerely thank you for your time in reviewing our paper, the reviews, and our responses. We completely understand the huge burden you have in this decision period. Therefore, to help you quickly digest, in this cover letter, we concisely (1) summarize the contributions of our paper, and (2) highlight how we have addressed each concern raised by the reviewers.

### **1. Contributions**

Our paper proposes **IPGO**, *Indirect Prompt Gradient Optimization*, which is a **novel** and **model-agnostic framework** that aligns images produced by Text-to-Image diffusion models with external rewards, such as Semantic Alignment (CLIP), Human Preference, and Aesthetics. Specifically, the main contributions are:

a. ***A novel T2I alignment framework – Constrained Text Embedding Insertion***: IPGO inserts rotated reduced-rank prefix and suffix embeddings at the beginning and the end of the original prompt embeddings. These inserted embeddings are learned jointly under range constraints, orthonormality constraints, and a conformity penalty, allowing the model to adjust alignment without modifying the underlying T2I model.

b. ***Two strategies to mitigate reward hacking***: in-training – increasing the conformity penalty; and post-training – blending Aesthetics/CLIP prefix–suffix embeddings.

c. ***Extensive computational experiments***: We demonstrate the effectiveness of IPGO against **6 strong SOTA baselines** across **3 datasets**. Our ablation studies systematically examine
- (i) the effectiveness of the constraints and the penalty,
- (ii) the impact of the prefix and suffix lengths,
- (iii) how these lengths interact with the original prompt length, and
- (iv) IPGO’s robustness across different diffusion models (SD3, SDXL and FLUX).


### **2. Reviewers’ Major Concerns and Our Responses**
a. Investigating and mitigating reward hacking in IPGO – **Done**

b. Comparing IPGO against noise optimization baselines, such as Best-of-N - **Done**

c. Justifying and testing the range and conformity constraints used in IPGO – **Done**

d. Showing IPGO can adapt to long prompts – **Done**

e. Applying to other large diffusion models, e.g., FLUX, SD3, and SDXL – **Done**

f. Fixing terminology (single-prompt optimization, reduced-rank) and typos – **Done**

*We have heeded all comments raised by the reviewers*, and the details of these changes are included in the revised manuscript. We are very grateful for all thoughtful comments and questions, which have helped to significantly improve our paper. We hope that you will find the paper acceptable for ICLR.

Thank you very much for your consideration!

Best regards,

The Authors

---

### Meta-Review · Area_Chair_RX3c · 2026-01-07

**Summary:**

Initial scores were mixed and leaned negative. Most concerns were about the experimental results and whether the method is useful in practice. Reviewer gMyg noted that IPGO is a costly test time optimization method, yet it is compared against training based baselines such as TextCraftor that run with instant inference. In the rebuttal, the authors state IPGO needs 3 to 5 minutes of optimization per image, which makes that comparison unfair. Reviewer FprH also highlights that using only a prefix or only a suffix performs almost the same as the full prefix plus suffix setup, 0.289 versus 0.290, so the main design choice is hard to justify. The rebuttal does not resolve these key issues, so there is no basis to accept. We encourage the authors to take into account the reviewers' comments and to resubmit to another venue.

**Reviewer Concerns:**

Addressed:
Generalization: aZrb and FprH. The rebuttal adds results on SDXL, SD3, and FLUX 1.0 Schnell, so the method is not tied to SD1.5.

Math detail: fnYW. They fixed the dimension mismatch in Equation 3.

Long prompts: aZrb. They added results on prompts longer than 150 words using FLUX.

Outstanding:
Runtime and fairness: gMyg. IPGO still takes 3 to 5 minutes per image, so comparisons to instant inference methods remain misleading.

Reward hacking: gMyg. On aesthetic rewards the method can increase the score and semantic alignment drops. The proposed fixes read like patches, not a stable solution.

Design redundancy: FprH. Prefix only or suffix only is almost the same as the full model, 0.289 versus 0.290.

**Reviewer Scores:**

gMyg: stays at 2. The rebuttal confirms the long runtime per image, and the gains over Best of N are not enough to justify it.

aZrb: stays at 4. Generalization is clearer now, but the method still looks too slow per image.

FprH: likely stays 6. The ablation suggests the prefix plus suffix design is redundant.

fnYW: stays at 4. The equation fix helps, but it does not change the main concerns.

---

### Decision · Program_Chairs · 2026-01-26

Reject